

# Model sensitivity across scales: a case study of simulating an offshore low-level jet

Patrick Hawbecker[1], William Lassman[2], Timothy W. Juliano[1], Branko Kosović[3], and Sue Ellen Haupt[1]

[1]National Center for Atmospheric Research; 3450 Mitchell Ln, Boulder, CO 80301, USA
[2]Xcel Energy; work performed while at Lawrence Livermore National Laboratory, Livermore, CA 94550, USA
[3]Johns Hopkins University, Baltimore, MD 21218, USA

**Correspondence:** Patrick Hawbecker (hawbecker@ucar.edu)

**Abstract.** In this study, a seven-member ensemble of mesoscale-to-microscale simulations with varying sea surface temperature (SST) is conducted for a case in which an offshore low-level jet was observed via floating lidar. The performance of each SST setup in reproducing physical characteristics of the observed low-level jet is compared across the mesoscale and microscale domains. It is shown that the representation of low-level shear, jet nose height, and hub-height wind speed are improved when moving from mesoscale to microscale. Specifically, low-level shear is improved in the microscale by reducing near-surface wind speeds and lowering the jet nose height to be closer to that observed. Counterintuitively, the sensible heat flux on the mesoscale domains is more negative than on the microscale domains, which would indicate a more stable boundary layer with higher shear; however, the low-level shear in the mesoscale is weaker than that of the microscale domains. This indicates over-mixing of the PBL schme in the mesoscale domains and/or over-prediction of surface drag in the microscale domain.

We analyze performance considering a real-world scenario in which the computational burden of running an ensemble of LES limits a study to performing a mesoscale ensemble to select the *best* model setup that will drive a single LES run. In the context of this study, the best model setup is subjective and weighs model performance in the physical representation of the low-level jet as well as the model surface forcing through the temperature gradient between air and sea. The expectation of this approach is that the best performing setup of the mesoscale simulations will produce the best result for the microscale simulations. It is shown that there are large fundamental changes in the characteristics of the low-level jet as well as in the surface forcing conditions between the mesoscale and microscale domains. This results in a non-linear ranking of performance between the mesoscale domains and the microscale domains. While the best performing mesoscale setup is also deemed to produce the best results on the microscale, the second best performing mesoscale setup produces the worst results on the microscale.

## 1 Introduction

Conducting large-eddy simulations (LES) driven by mesoscale numerical weather prediction (NWP) models for real data cases has become increasingly popular for meteorological and wind energy related studies. With advances in computational power, it is increasingly manageable to run LES on large domains (e.g., tens of kilometers) down to fine scales (e.g., several meters) in





order to simulate complex flows and atmospheric phenomena of interest. However, studies running simulations in this manner are often limited to a single LES run due to the computational burden. In real data cases, domain sizes often are quite large while model grid spacing, $\Delta_x$, is in the 10's of meters if not smaller. LES domains must be sufficiently large in size in order to capture the meteorological event of interest. Additionally, periodicity is not an appropriate boundary condition for real data cases because initial and boundary conditions must be specified and allowed to vary in time. In these cases the boundary

conditions for the LES are typically derived from NWP models at mesoscale resolution. When nesting down to LES scales the model is tasked with filling in the energy, or turbulence, at scales not resolved by the mesoscale model at the boundaries. This requires time and distance for turbulence to develop known as "fetch" (Muñoz-Esparza et al., 2014; Mirocha et al., 2014; Haupt et al., 2019, 2020). While there are techniques to decrease the turbulent fetch region, such as the stochastic cell perturbation technique used in this study (Muñoz-Esparza et al., 2014, 2015), they still require a considerable amount of the domain to

be dedicated strictly to developing adequate turbulence. From this, boundary-coupled mesoscale-to-LES simulations often demand a large domain size for the LES while retaining a small grid spacing. The LES domain will also require time to spin-up turbulence before the flow field can be analyzed. Thus, running such simulations is often computationally expensive.

       For case studies in which obtaining an accurate representation of the flow field at turbulence-resolving scales from LES is required, minimizing computational cost can be challenging. One approach is to run several mesoscale simulations (which are

relatively inexpensive, computationally) to find the best performing *mesoscale* setup and then use that setup to drive the LES. The assumption in this approach is that the domain–averaged LES solution will not differ largely from the mesoscale result, and that the sensitivity on the mesoscale will directly translate to the microscale. Thus, the best performing mesoscale setup is presumed to lead to the best performing microscale setup.

       In this study, model sensitivity of an offshore low-level jet (LLJ) to sea-surface temperature (SST) is analyzed across both

the mesoscale and microscale. The goal is to assess the sensitivity of LLJ characteristics and performance when compared to observations in order to determine whether the assumptions above are indeed valid.

       The parameter choices for this sensitivity study are twofold. First, model results are likely sensitive to many factors such as turbulence closure, surface layer parameterization, initial and boundary conditions, etc. When adjusting parameterizations such as the planetary boundary layer (PBL) scheme on the mesoscale domain, there is no direct corollary for these PBL

schemes on the microscale. In Talbot et al. (2012), sensitivity to the PBL scheme was assessed on the mesoscale domain along with the sub-grid scale turbulence closure scheme on the microscale domain. While the PBL scheme and sub-grid turbulence closure schemes have similar functions, they are not directly comparable across scales. Thus, it is difficult to assess how model sensitivity changes across scales when the mesoscale and microscale domains are each run with varying turbulence closure techniques. Additionally, other parameterizations within the model are scale-sensitive and their impacts vary as $\Delta_x$ changes.

Thus, it is important to select a parameter that will impact the mesoscale and microscale domains similarly to assess how sensitivity changes across scales.

       A second consideration for parameter choice is to provide a way to vary the characteristics of the LLJ. As offshore wind energy continues to grow in the United States, observations have shown that LLJs are a common occurrence off the coast of the Mid-Atlantic and Northeast states (Zhang et al., 2006; Colle and Novak, 2010; Nunalee and Basu, 2014; Colle et al., 2016;



Pichugina et al., 2017; Strobach et al., 2018; Debnath et al., 2021; Aird et al., 2022; De Jong et al., 2024; Quint et al., 2025) and their impacts on energy production and turbine performance must be considered (McCabe and Freedman, 2025; Paulsen et al., 2025). These LLJs have been shown to have jet noses frequently below 100 m, which would mean in the offshore environment, where turbines are larger than those on land, the shear profile across the rotor could be very complex. The presence of negative shear over the rotor swept is shown to impact turbine loads and stresses and decrease wake recovery (Wharton and Lundquist,

2012; Bhaganagar and Debnath, 2014; Park et al., 2014; Gutierrez et al., 2017; Kalverla et al., 2019; Gadde and Stevens, 2021). Debnath et al. (2021) showed relationships between the temperature difference ($\Delta T$) between air (2 m temperature) and sea (SST) and the occurrence and strength of strong shear or LLJ events in the New York Bight. Thus, any changes in $\Delta T$ are likely to augment the characteristics of the LLJ. We elect to vary SST in this study as a simple way to augment $\Delta T$ that is consistent across all domains.

Preliminary results of this study have been presented (Hawbecker et al., 2022) and a brief summary of the project and high-level conclusions have been shared in Haupt et al. (2023). Here, we share in more detail the full analysis of this study as well as additional conclusions pertaining to the differences in mesoscale and LES performance, surface characteristics and their impact of the simulated LLJ, and important requirements for running WRF at high resolution (O(1-10)m). Figures that have been reproduced from Haupt et al. (2023) are noted in the respective captions.

Section 2 discusses the observational dataset and various SST products that are included in this study. The model setup and computational discussion can be found in Section 3. Results are shown in Section 4, followed by a summary and discussion in Section 5.

## 2   Data and Methods

The modeled sensitivity of LLJ characteristics to SST is examined by including several auxiliary datasets. These datasets,
which are produced from different satellite sensors, vary in spatial resolution and are described below.

### 2.1   Sea-Surface Temperature Datasets

The SST datasets used in this study are derived from varying underlying instrumentation, and are available at varying spatial and temporal resolution (Figure 1 and Table 1). Five SST datasets are downloaded from the Group for High Resolution Sea Surface Temperature Level-4 (GHRSST-L4) database including: the Canadian Meteorological Center (CMC) analysis product (Canada
Meteorological Center, 2017), the Office of Satellite and Product Operations (OSPO) analysis (OSPO, 2015), the Multiscale Ultrahigh Resolution (MUR) dataset (NASA/JPL, 2015), the Naval Oceanographic Office (NAVO) dataset (NASA Jet Propulsion Laboratory, 2018), and the Operational Sea Surface Temperature and Sea Ice Analysis (OSTIA) analysis (UKMO, 2005). Additionally, an SST dataset from the Level-3 GOES-16 Advanced Baseline Imager (GOES-16) is downloaded from the GHRSST database (NOAA/NESDIS/STAR, 2019) and gap-filled using Data Interpolating Empirical Orthogonal Functions
(DINEOF) (Beckers and Rixen, 2003; Alvera-Azcárate et al., 2005; Beckers et al., 2006; Alvera-Azcárate et al., 2007).





Initial and boundary conditions for model in this study are derived from the MERRA-2 reanalysis dataset (Global Modeling and Assimilation Office (GMAO), 2015e). Within the MERRA-2 dataset, SST comes from the OSTIA dataset (Figure 1a). Each auxiliary dataset overwrites skin temperature over water within the initial and boundary conditions for each simulation (Figure 1b–g). On the outermost domain for the GOES-16 simulations (Figure 1f), skin temperature is overwritten only over

the area covered by domain-2 due to processing constraints. Because the spatial extent of domain-2 is large (600 km × 600 km) and encompasses the region of the ocean over which winds in this study are coming from, we expect this to have a negligible impact on the flow field over the region of interest. For every other setup, the satellite-derived SST products overwrite the skin temperature over water for the entirety of each domain.

Differences in the resolution of features between the various SST datasets is apparent (Figure 1). Most notably, the gradients

in SST, which can be important forcing mechanisms for offshore LLJ formation (Gerber et al., 1989; Small et al., 2008; Whyte et al., 2008), are better captured at finer scales in the GOES-16 and OSTIA datasets (Figure 1f and e, respectively) than in the lower-granularity datasets such as NAVO and CMC (Figure 1b and c, respectively). OSPO (Figure 1d) and OSTIA (Figure 1e) share the same granularity, yet the features within the OSTIA dataset contain much finer detail. The MUR dataset has the finest granularity (Figure 1g) yet the resulting product appears much smoother than that of GOES-16 (twice the granularity) and even

OSTIA (more than five-times the granularity). Note that while the native SST dataset within the MERRA-2 reanalysis product (Figure 1a) is produced from the OSTIA dataset (Figure 1e), the resolution of the MERRA-2 product is at the resolution of the MERRA-2 data at 0.62° latitude and 0.5° longitude. Based on the differences between these products, we expect to see differences in the simulated characteristics of the offshore LLJ such as in jet-nose height, maximum wind speed, and low-level shear.





**Figure 1.** Sea surface temperature over domain 2 for each SST dataset. The locations of the E06 buoy (filled) and E05 buoy (open) are designated by an "X" in each panel. The outline of domain 3 is also shown with a dotted line. This figure has been redrawn to include domain 3 from Figure 8 in Haupt et al. (2023).

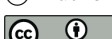



**Table 1.** Spatial granularity and types of sensors used in each WST dataset.

| WST Dataset | Spatial Granularity (Lat. and Lon.) | Sensors |
|---|---|---|
| NAVO | 0.1° | 5 satellite sensors |
| CMC | 0.1° | 7 satellite sensors, drifting/moored buoys |
| OSPO | 0.054° | 6 satellite sensors, ships, drifting/moored buoys |
| OSTIA | 0.054° | 7 satellite sensors, drifting/moored buoys |
| GOES-16 | 0.02° | 7 satellite sensors, drifting/moored buoys |
| MUR | 0.01° | 6 satellite sensors, drifting/moored buoys |

## 2.2 Observations

In 2019, two EOLOS FLS200 floating lidar buoys were deployed in the New York bight by the New York State Energy Research and Development Authority (NYSERDA; OceanTech Services/DNV under contract to NYSERDA (OceanTech Services/DNV, 2019)). These buoys, named E05 and E06 (open and filled *X* in Figure 1, respectively), contain vertical scanning lidars, ocean and wave sensors, and a small meteorological mast. Each lidar records data at 10 levels from 20 m to 200 m above mean sea level at 20 m increments. The data are corrected for tidal variation and freely available at 10-minute averaged output. While both E05 and E06 capture the event of focus, only data from E06 is considered within this study as explained in more detail in Section 3.

## 2.3 Low-level jet case

The case study of interest in this work consists of an offshore LLJ that developed in the evening of April 5, 2020 off the coast of the Mid-Atlantic states, USA (data from the E06 buoy is shown in Figure 2). The jet begins as 2 m air temperature begins to rise and SST slowly decreases. The difference in 2 m air temperature and SST, $\Delta T$, serves as a proxy for atmospheric stability, where when positive one can expect stable atmospheric conditions. As the air and sea surface temperatures diverge and $\Delta T$ grows larger, a strengthening in wind speeds occurs around 100 m and eventually leads to the formation of an LLJ. Over the six-hour period of interest beginning at 00Z on April 6 (8 PM local time on April 5), the jet nose remains at ∼120 m at the E06 buoy location with maximum wind speeds around 16 m s$^{-1}$. The LLJ persisted for several hours before weakening as $\Delta T$ decreased towards zero with the passing of a cold front.




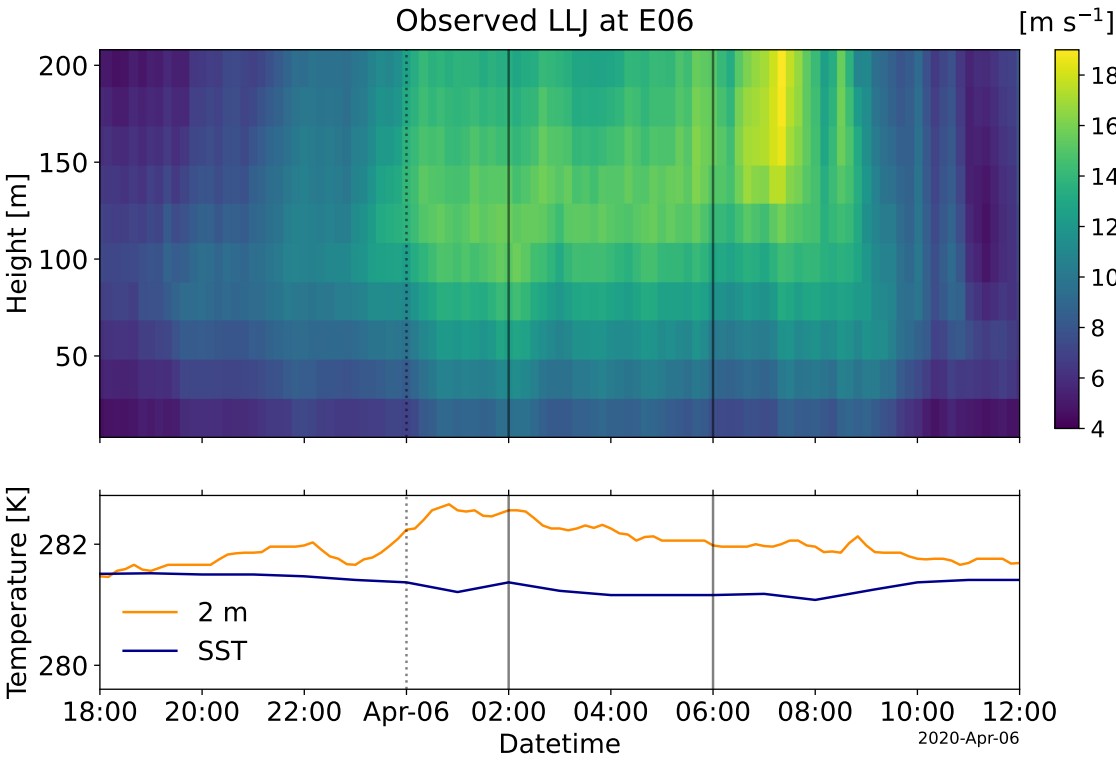

**Figure 2.** Observed wind speed from the E06 lidar (a) and 2 m air temperature and SST from the E06 buoy (b). Dotted vertical lines are the start of the LES simulations and solid vertical lines denote the period of interest.

## 3 Model Setup

Simulations in this study were conducted using the Weather Research and Forecasting (WRF) model version 4.3 (Skamarock et al., 2021). Each simulation comprised five one-way nested domains with model grid spacing, $\Delta_x$, set to 6,250 m, 1,250 m,
250 m, 50 m, and 10 m on domains 1 through 5, respectively (Figure 3). LES domains 4 and 5 are positioned with the E06 buoy in the northeast portion of the domain so that the incoming southwesterly flow has ample space to develop turbulence. Due to the spacing between the two floating lidars, it would be very difficult to run simulations with LES domains spanning both lidars. Thus, in this study, we focus strictly on data from the E06 lidar.

Model time step on domain 1 is set to 15 seconds and we use a time-step ratio of 3, 5, 5, and 5, for each nest. The vertical grid
contains 131 levels with 21 levels below 200 m. The same vertical grid is defined for each domain such that vertical resolution is held relatively constant between each domain. Each domain also shares the following parameterizations: the revised Monin-Obukhov surface layer scheme (Jiménez et al., 2012), Ferrier microphysics (Ferrier, 2004), RRTMG longwave and shortwave radiation (Iacono et al., 2008), and the unified Noah surface layer model (Tewari et al., 2004). Domain-1 uses the Kain-Fritsch cumulus parameterization (Kain, 2004) and four-dimensional data assimilation (Liu et al., 2005, 2007; Reen and Stauffer,



2010). Turbulence closure on domains 1 and 2 is performed with the MYNN 2.5 planetary boundary layer scheme (Janjić, 2002). These domains will be referred to as the *mesoscale* domains. Domains 3–5 utilize the 1.5 order SGS turbulent kinetic energy (TKE) scheme (Deardorff, 1980) and are considered the *microscale* domains.

Simulations are initialized with the MERRA-2 reanalysis dataset as initial and boundary conditions (Global Modeling and Assimilation Office (GMAO), 2015a, b, c, d, e, f) at 06 UTC on April 4, 2020 and are run for 48 hours. At initialization, only
domains 1 and 2 begin. At 18 UTC on April 5, domain 3 initializes and the three domains are run for six hours. Finally, at 0 UTC on April 6, domains 4 and 5 initialize and the five-domain setup runs for an additional six hours (dotted black vertical line in Figure 2). We utilize the stochastic cell temperature perturbation method on all boundaries of domains 4 and 5 (Muñoz-Esparza et al., 2014, 2015) to accelerate turbulence development. The first hour of the simulations are considered spin up then the final five hours of simulation are considered for analysis (between the solid black vertical lines in Figure 2). Due to the
computational expense of these runs, each simulation must restart after 20 minutes of simulation time while all five domains are running. Each domain produces output every 10 minutes to match the output frequency of the observations. Pseudo-tower (the *tslist* option in WRF) output was also produced at the location of the E06 tower for higher-temporal resolution analysis. However, in running the simulations for this study, an issue within the WRF pre-processing system (WPS) was found that impacted the LES domains; specifically the COSALPHA and SINALPHA calculation (Figure A2) and corrupted the psuedo-
tower (tslist) output. This issue is explained in more detail in Appendix A along with an analysis of the impact on the model solution (Figure A1). Within this manuscript, we refer to a *setup* as all domains in a simulation for a single SST product.

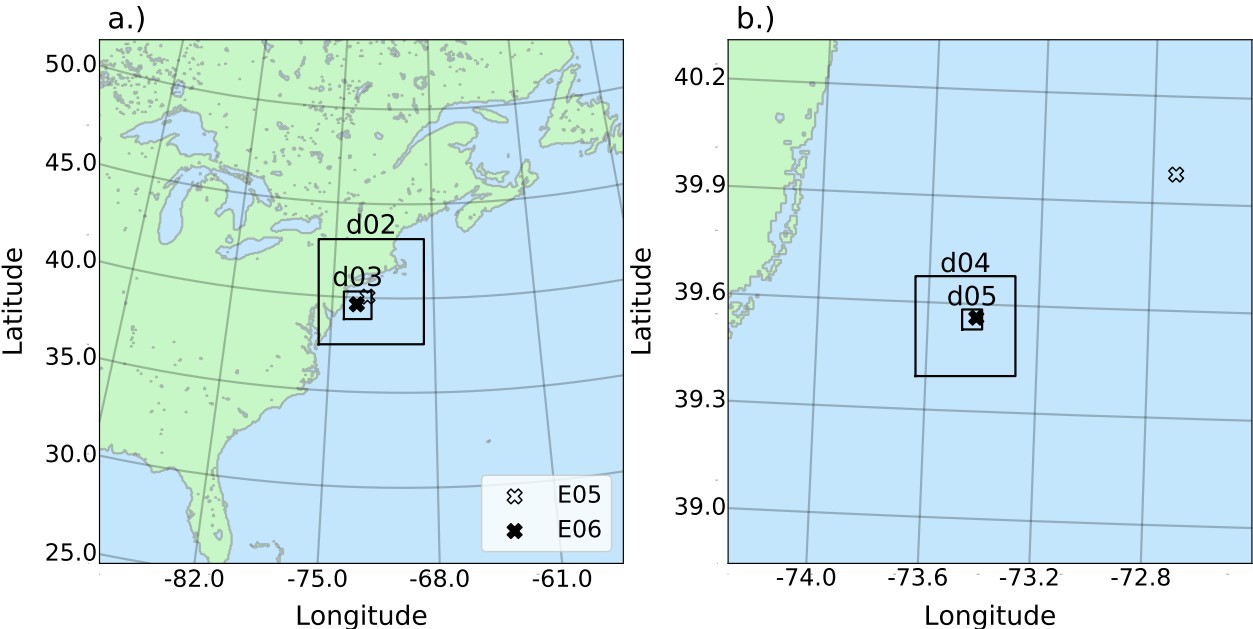

**Figure 3.** Domain configuration within WRF. The locations of the E06 buoy (filled) and E05 buoy (open) are designated by an "X" in each panel. The outer extent of (b) is the perimeter of domain 3.





## 4 Results

The following results are derived from each simulation at the location of the E06 buoy for all domains. For domains 1 and 2 – the mesoscale domains – data is extracted from the single cell that encompasses the E06 buoy location. For domains 3, 4, and 5 data is spatially averaged over a block of cells that are centered over the E06 buoy location and cover the footprint of one domain-2 cell. This is done to average the turbulent results on the microscale in order to compare the mesoscale and LES results more faithfully. While domains 4 and 5 are considered LES or microscale domains, domain 3 has a grid spacing of 250 m, which is within the *terra incognita* (Wyngaard, 2004) or gray zone. LES turbulence closer techniques are used in this study within this domain though the appropriateness of this modeling technique is questionable. The other option is to use a planetary boundary layer scheme at this resolution, but the applicability of such parameterizations at 250 m grid spacing are even more questionable. An investigation of the applicability of a three-dimensional PBL scheme in WRF (Kosović et al., 2020; Juliano et al., 2022; Eghdami et al., 2022) within this domain may shine light on using such a scheme as a potential alternative moving forward.

Analysis focuses on the representation of the wind field and characteristics of the simulated jet when augmenting the SST dataset (as discussed in Section 2) that is ingested into WRF. The observations in this study are limited to 10-minute averaged wind profiles over time, measurements of temperature at 2 m, and SST. The offshore wind turbine specifications assumed in this study consist of a hub-height of 118 m with a rotor diameter of 160 m. These dimensions are similar to typical offshore wind turbines currently installed as of the writing of this paper (Stehly and Duffy, 2021; Musial et al., 2023). Analysis of SST and low-level jet characteristics is performed first. Ensemble statistics are then calculated in order to determine the sensitivity of the jet characteristics to SST and how that sensitivity changes across domains. Lastly, it is determined whether selecting the single best-performing mesoscale setup for driving LES will provide the best solution on the microscale for this low-level jet case. It is important to note that one could run a suite of model configurations for the microscale domain for each SST setup in order to improve the microscale model simulation associated with each mesoscale simulation. In practice, however, the LES simulations are very computationally expensive to run as previously mentioned. Thus, in this study, we select a single LES configuration and run it for each SST setup – for better or for worse – to emulate a real-world scenario in which only a mesoscale suite of simulations is run that is used to determine the best single SST setup to drive an LES simulation.

### 4.1 SST Depiction

For each simulation, the average modeled SST value at the E06 buoy may vary from domain to domain (Figure 4). This is more apparent when the higher-resolution SST datasets (GOES-16 and MUR) are employed. For the lower-resolution domains and default SST dataset, SST may differ between domain 1 and domain 2, but values for domains 3-5 are very similar. Note that each dataset depicts colder SST values than what was observed.





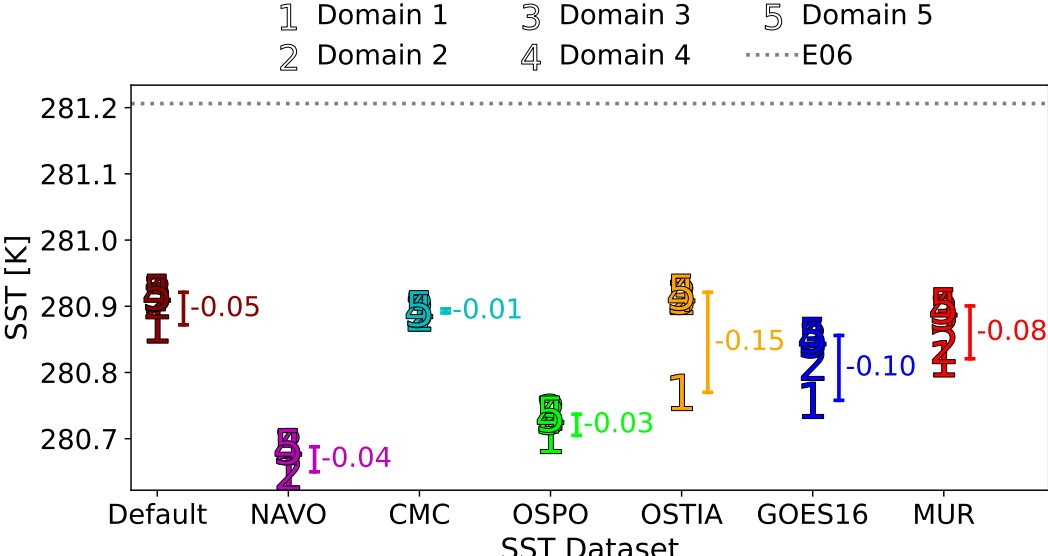

**Figure 4.** Average SST for each setup and each domain along with the difference between SST on domain 1 and domain 5.

## 4.2 Low-level Jet Characteristics

Low-level jets have historically been detected via a vertical profile of wind speed in which a maximum is reached at some height above the surface followed by a decrease in wind speed above that level. For wind energy purposes, shear – the difference in wind speed over a certain height, $dU/dz$ – is often considered over the rotor swept area. In this study, we will consider a rotor swept area from 38 m to 198 m (solid black lines in Figure 5). We define low-level shear as shear between the bottom of the rotor swept area (38 m) and hub-height (118 m; dashed line in Figure 5).

During the period of interest, a wind speed maxima between 80 and 150 m is observed (Figure 5a). On the mesoscale domains (Figure 5b and c), a maximum wind speed is reached around 200 m. (There is a decrease in wind speed above this height in the mesoscale domains confirming that this is a low-level jet profile; not shown.) At low levels, the mesoscale simulations (Figure 5b and c) often produce higher wind speeds than in observations (Figure 5a). Here, low-level shear within the mesoscale simulations is too weak. The gray zone and microscale domains (Figure 5d–f) recover the jet profile and bring the jet maxima down to levels near, but slightly higher than in observations. The low-level wind speeds are much closer to observations, but with a stronger jet maximum wind speed resulting in stronger shear than than observed. In each of the simulations, the LLJ peaks around 160 minutes before seen in the observations (see Figure 2 for comparison). When shifting the observations to better match with simulated results (not shown), some metrics are slightly improved but the overall message remains unchanged. The same can be said for changing the period of interest to several 3–5 hour intervals between 01 UTC and 06 UTC (not shown).

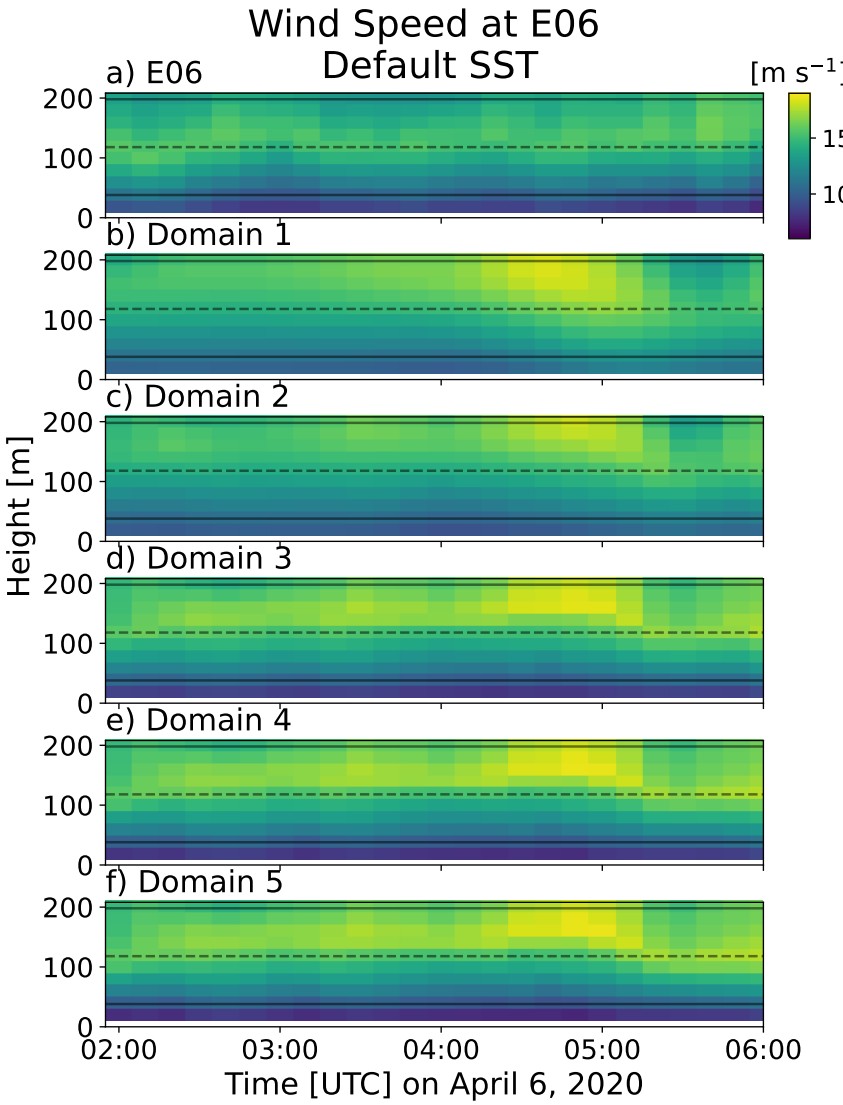

**Figure 5.** Vertical profiles of wind speed with time for observations (a), and domains 1, 2, 3, 4, and 5 (b–f, respectively) for the Default SST setup during the period of interest. The assumed hub height (dashed line) and rotor layer (between the solid lines) in this study are also shown.

Comparing hub-height wind speeds for each SST setup (Figure 6) displays some subtle variability as SST is augmented 205 between the datasets, but significant variation exists between domains within a setup. The differences between the mesoscale domains (shades of red) is larger than between the two LES domains (shades of blue), which follow each other closely. Domain 3 hub-height wind speeds fall in between that of domains 2 and 4 but more closely resemble the LES solutions. The gray zone and LES domains generally improve the simulation of hub-height wind speeds for much of the period of interest. Towards 05 UTC these wind speeds increase to well above what was observed. The mesoscale domains are deficient in hub-




height winds for the majority of the period but increase to around what was observed at 05 UTC. During this ramp up in wind
speeds, domain 1 wind speeds often increase well beyond those observed while domain 2 wind speeds remain much closer to
the lidar wind speeds. By and large, we see improvement in hub-height wind speed predictions as grid spacing increases with
jumps in performance occurring between the mesoscale domains and when transitioning from mesoscale to LES on domain 3.

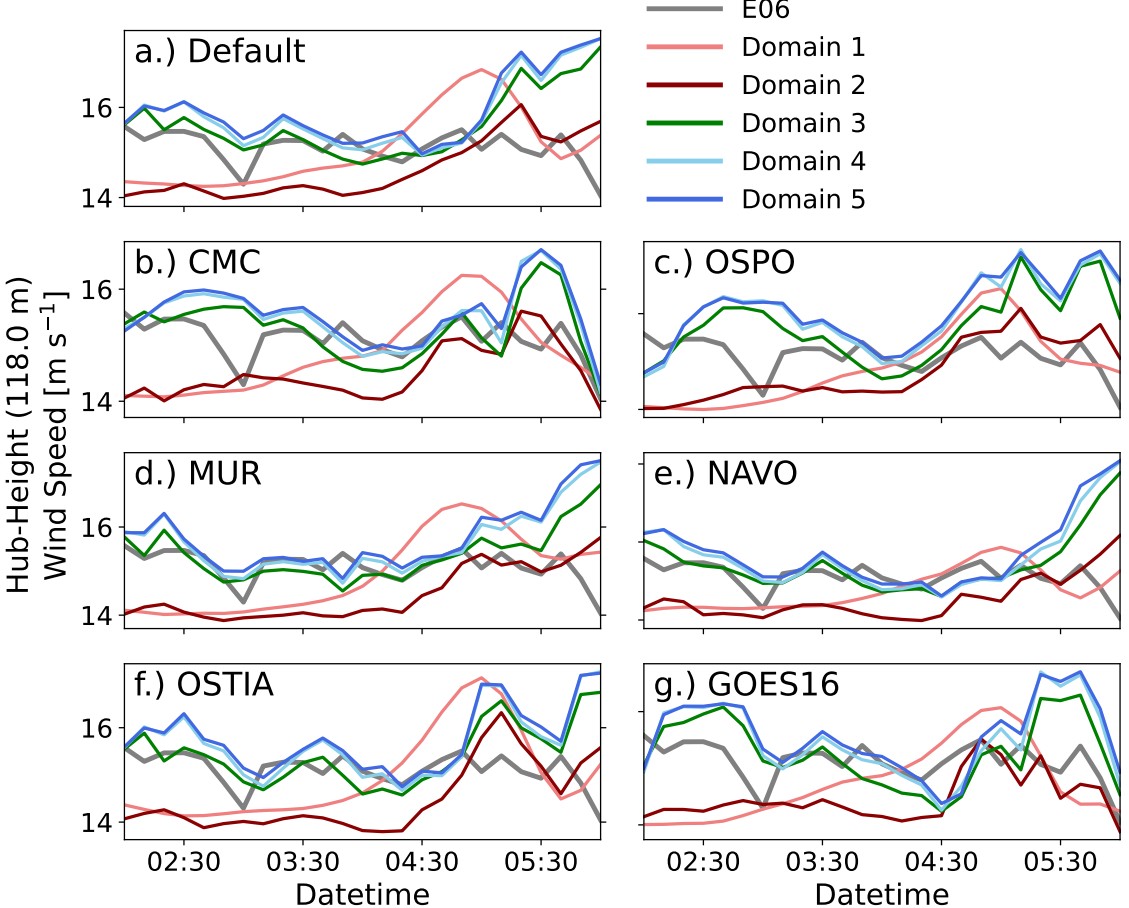

**Figure 6.** Hub-height wind speed for the mesoscale (shades of red), gray zone (yellow), and microscale (shades of blue) domains for each
SST setup along with the observed hub-height wind speed (grey).

Low-level shear generally improves when moving from the mesoscale domains to LES with, again, a large jump in per-
formance on domain 3 (Figure 7). The mesoscale domains under-predict the low-level shear while LES domain predictions
come closer to the observed shear, but are too large. The gray-zone results are closest to observations in all SST setups. For
the performance on the LES domains, this is mostly due to wind speeds being too low at the bottom of the rotor layer. For the
mesoscale domains, wind speeds at the bottom of the rotor layer are faster than observed while wind speeds at hub-height are
too slow. Domain 3 benefits from slightly faster wind speeds than LES at lower levels, but similar wind speeds at hub-height,





which produces more accurate predictions of low-level shear. Again, the variability across the mesoscale domains is larger than between the microscale domains.

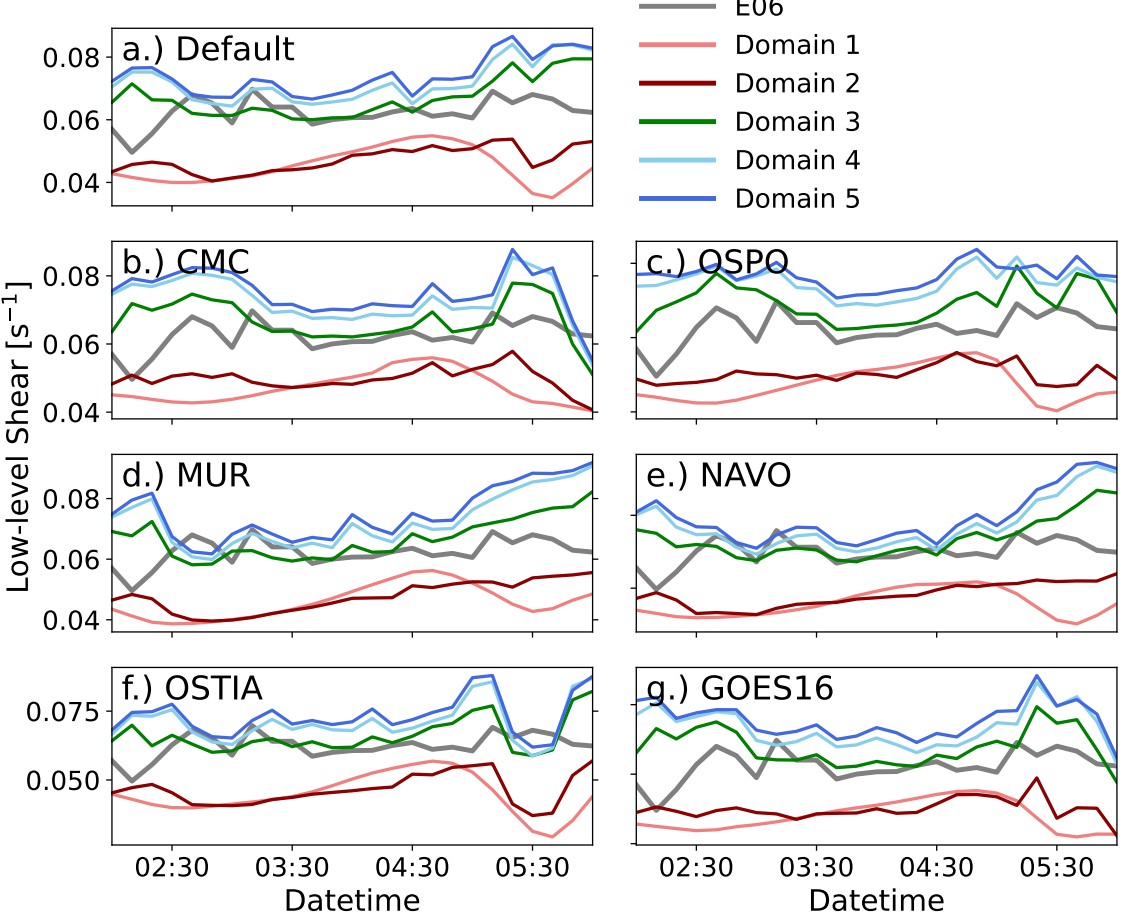

**Figure 7.** Same as Figure 6 but for low-level wind shear.

### 4.3 Ensemble Statistics

In order to quantify model sensitivity to SST across scales, it is helpful to consider results from all SST setups together as opposed to on a setup-by-setup basis. Recall that "setups" refers to the set of simulations run with varied SST datasets. Ensem-

ble results are generated on each domain by averaging data from all setups. Here we analyze the ensemble mean of individual variables and metrics (denoted by angle brackets), spread (Equation 1, where $N$ is the number of ensemble members), and ensemble mean error (EME; Equation 2) every 10 minutes. Note that the purpose of this ensemble is not to estimate the error, so we do not expect the values of these metrics to be similar.



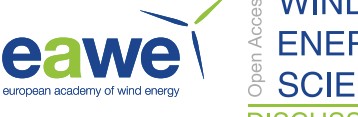

$$Spread = \sqrt{\frac{1}{N}\sum_{j=1}^{N}(member_j - Ens.Mean)^2} \qquad (1)$$

$$EME = \sqrt{(Obs. - Ens.Mean)^2} \qquad (2)$$

The vertical profile of the ensemble mean wind speed is drastically different between the mesoscale domains and LES domains over time (Figure 8a–c). On average, the LES domains slightly under-predict wind speed at low levels and over-predict the jet height and jet maximum wind speed (Figure 8d). Meanwhile, the mesoscale domains produce lower amounts of vertical shear and over-predictions of jet nose height. Further, the range in values over all SST setups on the LES domains

is often much larger than that on the mesoscale domains (Figure 8a–c). This produces a larger time-averaged spread (denoted by an over-bar) throughout the lowest 200 m in the LES domains when SST is varied (Figure 8e). Notably, domain 3 has a similar spread to domains 4 and 5 at upper levels, but produces spread between the mesoscale simulations and remaining LES simulations below the jet nose. Increasing the model resolution from domain 1 to domain 2 results in an increase in spread of wind speed throughout the majority of the profile.

EME, averaged in time, is reduced near the surface as resolution increases from domain 1 through to domain 3, but then begins to increase again from domain 4 to domain 5 (Figure 8f). Above the observed jet nose, EME on the LES domains increases rapidly. This increase in error is mostly attributed to an over-prediction of maximum wind speed with an over-prediction of jet height. The mesoscale domains do not represent the shear and jet height well, but produce lower error by not over-shooting wind speed and over-predicting the jet height.

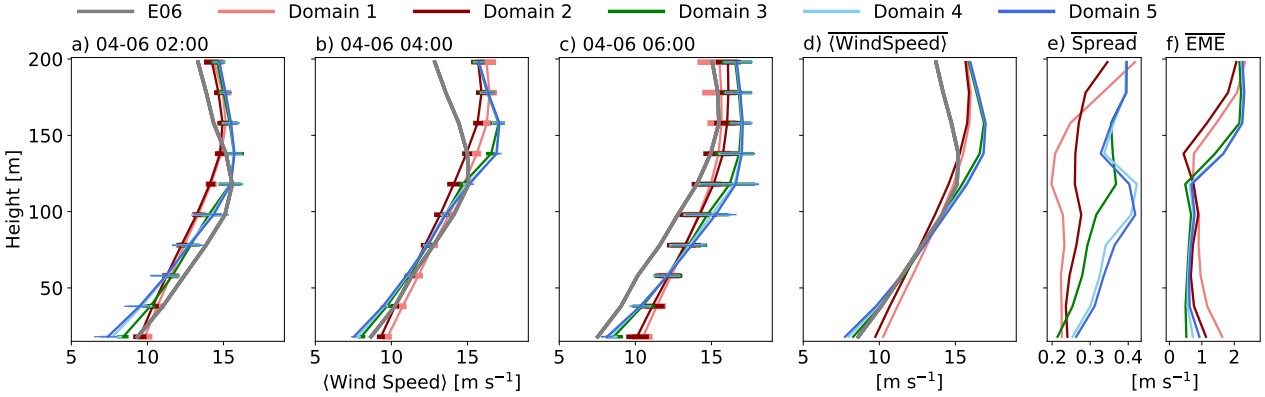

**Figure 8.** Vertical profiles of ensemble averaged wind speed at different times (a–c) for the mesoscale (shades of red), gray zone (yellow), and microscale (shades of blue) domains along with observations (grey). Error bars denote the spread between SST setups. The time-averaged ensemble average wind speed (d), spread (e), and ensemble mean error (EME; f) are also shown.





Changes in SST also generate larger spread on the gray zone and LES domains for low-level shear and hub-height wind speed (Figure 9a and c, respectively). Additionally, EME for these variables is also reduced as compared to the mesoscale domains for much of the period of interest on the gray zone and LES domains (Figure 9b and d, respectively).

Above hub height, spread increases for the mesoscale domains and error on the LES domains is higher than the mesoscale (Figure 8e and f). For this reason, we consider an integrated parameter over the rotor layer, the rotor equivalent wind speed (REWS), in order to determine the total impact of these variations within the environments considered important for wind energy. The calculation of REWS with veer used in this study is defined in equation (9) of Redfern et al. (2019):

$$REWS_d = \sum_{k=1}^{N} \frac{A_{ijk}}{A_T} |U_{ijk}| \cos(\theta_{ijk}) \tag{3}$$

where $N$ is the number of layers within the total rotor swept area, $A_T$. $A_{ijk}$ is the area of the rotor at a given level and $U_{ijk}$ is the average wind speed at this level. Veer is taken into consideration through $\theta_{ijk}$; the difference between the wind direction at hub height and the average wind direction in a given layer.

When considering the wind speed over the rotor swept area, spread on all domains is very similar (Figure 9e). EME of REWS is lowest on domain 2 and highest on the LES domains (Figure 9f). The mesoscale domains benefit from slightly under-predicting wind speeds below hub height and over-predicting wind speeds above (Figure 8d) which reduces error in REWS.



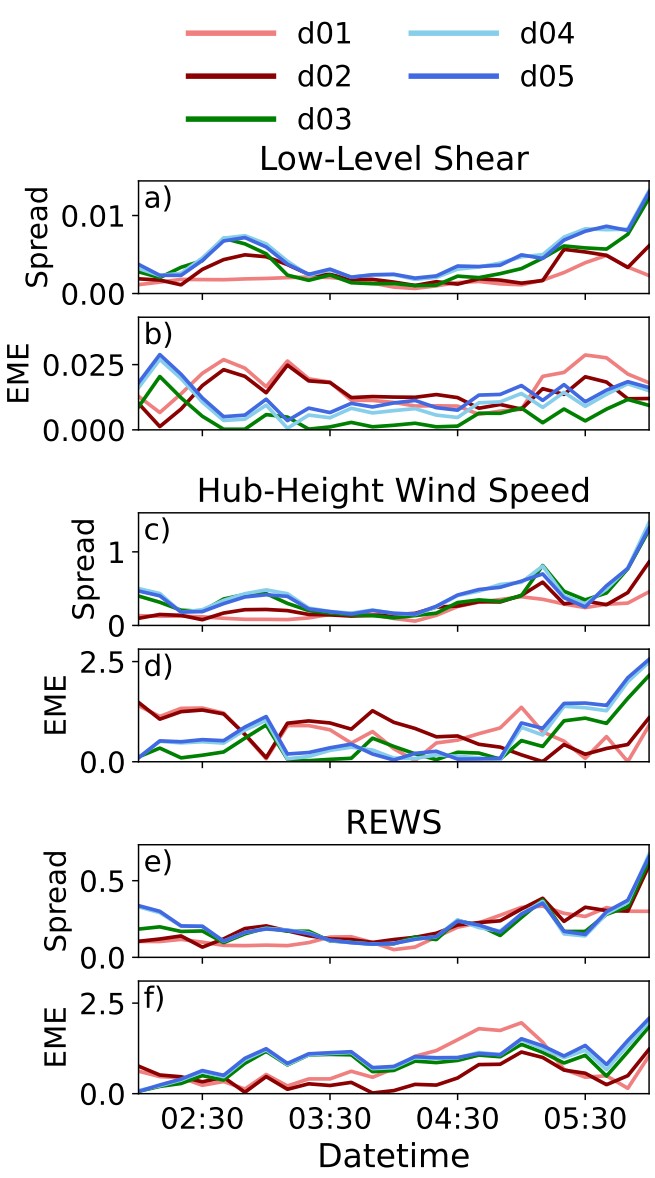

**Figure 9.** Spread and ensemble mean error on the mesoscale (shades of red), gray zone (yellow), and microscale (shades of blue) domains for low-level shear (a and b), hub-height wind speed (c and d), and rotor-equivalent wind speed (e and f).

Analyzing the time series of the ensemble mean of bias in low-level shear (Figure 10a) the mesoscale domains underpredict low-level shear while LES domains over-predict. The gray zone domain minimizes bias due to producing slightly faster near-surface wind speeds than the LES domains. In order to determine if increased stability is the cause of this disparity, the temperature difference between 2 m temperature and SST, $\Delta T$, is calculated (Figure 10b). While $\Delta T$ is not a perfect metric for stability, it is possible to compare the model against observations using this metric to glean some insight into the near-





surface stability. The bias for $\Delta T$ on each domain is similar until late in the period of interest. $\Delta T$ bias is predominantly near zero or positive indicating stronger stable conditions in the simulations. This is confirmed by examining surface sensible heat flux (SHFX; Figure 10c) where the values are negative throughout the period. It is interesting to note that the mesoscale domains produce larger negative values of SHFX than the LES domains, which indicates more stable conditions. In more stable conditions, one might expect shear to be stronger at low levels (Holtslag et al., 2014); this is not the case in these simulations.

For the majority of the period of interest, domain 1 produces a larger negative value of SHFX than domain 5 until around 0530 UTC when the SHFX values become similar. Inspecting low-level shear bias, the differences between domain 1 and domain 5 remain fairly constant throughout the period of interest. When the sudden increase in SHFX on domain 1 occurs (bringing the value close to that of domain 5), the difference in low-level shear bias is actually increased even though SHFX values become similar. This suggests that the separation in performance of low-level shear bias appears to be more closely

related to whether the domain is a mesoscale or microscale domain rather than the actual value of SHFX. It is possible that the PBL scheme on the mesoscale domains is over-mixing. It is also possible that the drag forcing over water on the LES domain is misrepresented, resulting shear being too strong at low levels. This is discussed in more detail in Section 5.

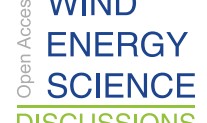

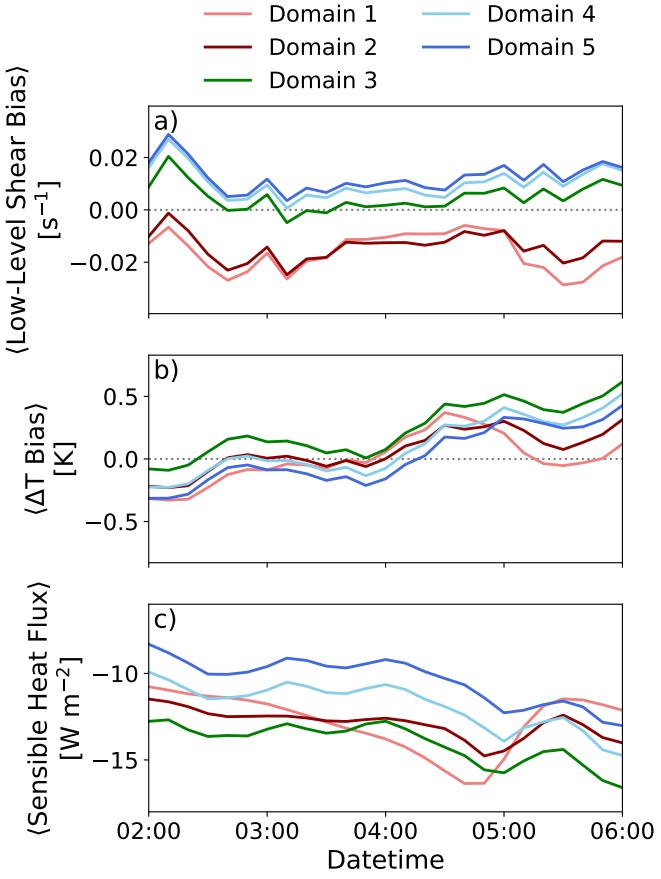

**Figure 10.** Ensemble means of low-level shear bias (a), $\Delta T$ bias (b), and surface sensible heat flux (c), for the mesoscale (shades of red), gray zone (yellow), and microscale (shades of blue) domains.

## 4.4 Predicting Performance Across Scales

Running an ensemble of LES is computationally expensive. To save computational resources and maximize model perfor-
mance, it is logical to perform various simulations varying components of the model (e.g., parameterizations, initial and boundary conditions, etc.) on the mesoscale in order to find the best model setup. This best setup can then be used to drive the LES run, which would be assumed to produce the best possible LES result from the available mesoscale setups. We recognize that the definition of "best" performer is subjective and likely to change based on the phenomena of interest as well as the metrics in which one is interested. Unless there is a single metric to be optimized, the comparison of simulation results requires
consideration of several variables and weighting the performance based on the interests of the study at hand. That said, when assuming that the best mesoscale result will produce the best microscale result, we are assuming that each LES simulation will perform similarly as the parent mesoscale simulation that drives it. This in turn assumes that the spread between the mesoscale runs and LES runs will be similar; the same goes for ensemble mean error. It has been shown that ensemble error and spread



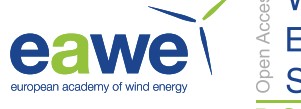

change among domains from the mesoscale to microscale, thus, we investigate whether we can safely assume that the best
performing setup on the mesoscale will lead to the best performing microscale simulation.

Considering root mean square error (RMSE) and bias for each setup on the mesoscale and microscale domains over the
period of interest for a variety of variables, model performance can be assessed to determine the top mesoscale performers.
The variables considered important in the context of wind energy here are: (1) low-level shear, (2) hub-height wind speed, (3)
REWS. On the mesoscale domains (domain 1 and domain 2), the lowest RMSE and smallest bias in low-level shear are from
the OSPO, GOES-16, and CMC SST datasets (Figure 11a and d). The same three datasets produce the lowest RMSE for hub-
height wind speed for domain 2, with only the NAVO dataset performing better on domain 1 (Figure 11b). Hub-height wind
speed is under-predicted on the mesoscale resulting in negative biases for all SST setups (Figure 11e). Considering REWS, the
CMC and GOES-16 datasets are among the top performers on the mesoscale domains, particularly domain 2, while OSPO is
among the worst performers for REWS in error and bias (Figure 11c and f). On the other hand, the OSTIA dataset is one of the
worst performers on the mesoscale domains for each of these metrics with the exception of REWS bias. Thus, the following
top performers for the mesoscale with respect to wind profile characteristics are identified as: GOES-16, CMC, and OSPO;
and the worst performing ensemble member is OSTIA (see Table 2).

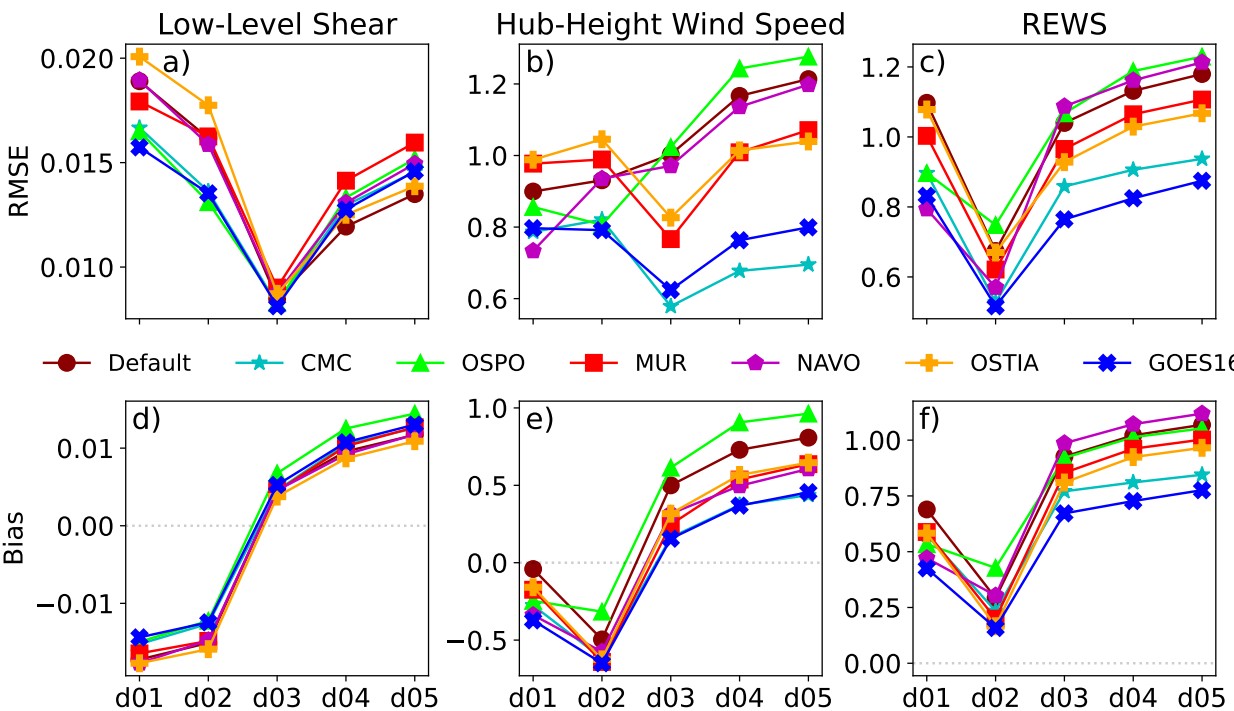

**Figure 11.** Root mean square error and bias for the mesoscale and microscale domains for all setups of low-level shear, hub-height wind speed, and REWS. This figure has been redrawn to include REWS from Figure 9 in Haupt et al. (2023).





To select a single best performing mesoscale setup, one might also consider how well the low-level forcing variables are captured in each setup. The low-level forcing variables considered here are: (1) 2 m Temperature, (2) SST, and (3) $\Delta T$). RMSE and bias of 2 m temperature at the E06 buoy location are best for the GOES-16 setup on the mesoscale domains while CMC and OSPO are in the mid-to-lower tier of performers (Figure 12a and d). For SST, CMC performs reasonably well while GOES-16 and OSPO are among the worst performers on the mesoscale domains (Figure 12b and e). The main driver for the surface forcing, however, is the difference between 2 m temperature and SST, $\Delta T$. The GOES-16 setup results in some of the highest error and highest bias (Figure 12c and f) leading to the assumption that perhaps the mesoscale domains in this case were getting the right answer for the wrong reasons. Meanwhile OSTIA, the worst performing ensemble member for the mesoscale domain, captures $\Delta T$ reasonably well with relatively low RMSE and the smallest bias. Of the three selected best mesoscale performers, CMC produces the best results for $\Delta T$ on the mesoscale and could reasonably be chosen as the setup to drive the LES runs. Ranking the performance is highly dependent on the specific feature being studied. For this context, we rank the mesoscale performers, from best to worst as follows: CMC, OSPO, GOES-16, Default SST, NAVO, MUR, OSTIA (Table 2).

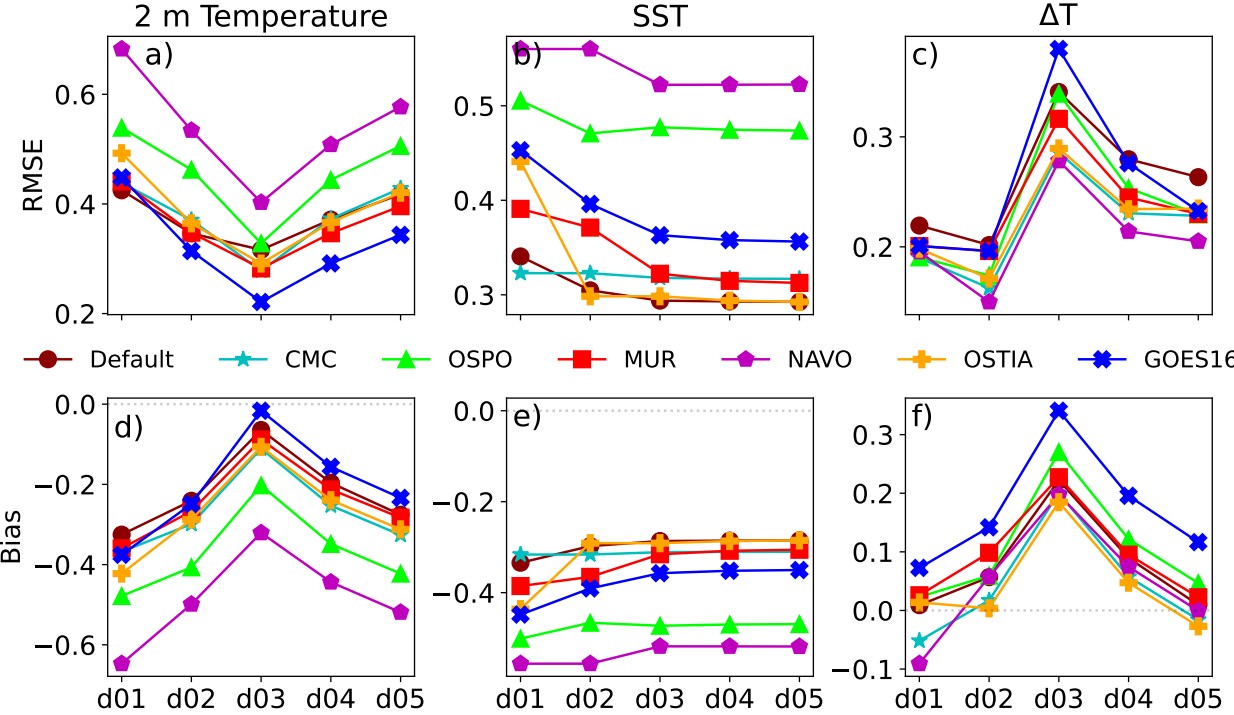

**Figure 12.** Same as Figure 11 but for 2 m temperature, SST, and $\Delta T$.

When the simulation grid spacing enters the gray zone with an LES turbulence closure scheme, the results for the wind field (Figure 11) and near-surface forcing (Figure 12) change drastically (with the exception of SST in which little variation is found across all domains). Biases for low-level wind shear and hub-height wind speed jump from negative to positive due to decreases in wind speed near the surface and higher wind speeds at the jet nose. RMSE of low-level shear reaches local





**Table 2.** Performance ranking of each setup for each domain.

|          | Domain 1 | Domain 2 | Domain 3 | Domain 4 | Domain 5 |
|----------|----------|----------|----------|----------|----------|
| Default  | 4        | 4        | 5        | 5        | 5        |
| CMC      | 1        | 1        | 1        | 1        | 1        |
| OSPO     | 2        | 2        | 6        | 7        | 7        |
| MUR      | 6        | 6        | 3        | 4        | 4        |
| NAVO     | 5        | 5        | 7        | 6        | 6        |
| OSTIA    | 7        | 7        | 4        | 3        | 3        |
| GOES16   | 3        | 3        | 2        | 2        | 2        |

minimum in all SST setups due to the near-surface wind speed being slightly faster than the LES domains while maintaining a
similar wind speed at hub-height (Figure 8d). Similarly, 2 m temperature reaches a local minimum on domain 3 while bias is
nearest to zero for all SST setups (Figure 12a and d).

Considering now the LES domains, the temperature forcing field's performance recover from the step change in domain 3
to remain more or less static between the mesoscale and microscale (Figure 12). There are some adjustments in the ranked
performance in these fields, but no large shifts from worst on mesoscale to best on microscale. From this, we might expect that
performance in the important variables identified above would also not change dramatically.

For the wind profile characteristic variables, the best performers in low-level shear on the LES domains are the default SST
setup and the OSTIA setup (Figure 11a and d). Recall that the OSTIA setup was the worst performer on the mesoscale. OSTIA
also improves to be among the middle-to-top performers for hub-height wind speed and REWS (Figure 11b, c, e and f). GOES-
16 and CMC performed at opposite ends of the spectrum for $\Delta T$ yet produce similarly good results on both the mesoscale and
microscale. Ranking the setups for microscale performance results in: CMC, GOES-16, OSTIA, MUR, Default SST, NAVO,
OSPO (Table 2). Note that two of the top three performers on the mesoscale are the top two of the microscale performance
ranking. However, OSPO moves from the second best overall performer on the mesoscale to the worst performer on the
microscale. Likewise, the worst mesoscale performer improves to the third best performer on the microscale domain.

OSPO was one of the better performers for low-level shear and hub-height wind speed on the mesoscale. Depending on the
metric that is most of interest to a study, it could have been selected as the best performing mesoscale study. However, the OSPO
setup is the worst performer for these metrics on the microscale (Figure 11a, b, d and e). This is significant due to the fact that
the SST and 2 m temperature (and resulting $\Delta T$) performance for OSPO remains fairly constant from mesoscale to microscale
– a finding that would suggest relative performance would also not change. The fact that a large swing in performance is found
suggests that there are other larger factors in determining model performance across scales.





## 5 Summary and Conclusions


Utilizing a suite of mesoscale-to-microscale WRF simulations of an offshore low-level jet in which we vary the SST dataset within the model, we analyze how modeled LLJ sensitivity to SST changes across scales. This sensitivity is analyzed based on physical properties of the simulated LLJ. We find that the mesoscale domains for each SST setup generally produce too little low-level shear and under-predict the hub-height wind speed. Conversely, the LES domains over-predict both low-level shear

and hub-height wind speed. The point at which the simulation shifts from underpredicting to overpredicting low-level shear and hub-height wind speed is on domain 3 – the domain within the *terra incognita* or *gray zone*. We find more variation between the mesoscale domains (domains 1 and 2) for each metric than between the LES domains (domains 4 and 5). Analyzing ensemble statistics for the jet characteristics, we find that ensemble spread in the LES domains is generally higher than the mesoscale domains and error is lower, specifically in the lower half of the rotor layer for this case. When we consider an integrated

wind parameter, rotor-equivalent wind speed (REWS), we find that model performance between LES and mesoscale is more comparable. Mesoscale domain 2 REWS results are shown to out-perform LES in ensemble mean error for the majority of the period of interest.

One of the main discrepancies between the ensemble average of the mesoscale and microscale domains was in the near-surface wind speed and resulting low-level shear. The microscale domains consistently simulated weaker low-level winds

resulting in being positively biased for shear below hub height while the mesoscale domains produced faster low-level winds leading to being consistently negatively biased. However, the mesoscale domains produced larger-negative sensible heat flux values during the period of interest. One would expect from this that the mesoscale domains show a higher level of stability than do the microscale domains, which would in turn produce *more* shear on the mesoscale domains. This finding leads us to believe that the mesoscale MYNN 2.5 PBL parameterization may overly mix the stable boundary layer, and/or the 1.5 order

TKE sub-grid turbulence scheme on the LES domains misrepresents surface drag over the ocean. Within WRF, the surface layer parameterization calculates the surface drag coefficient based on the assumption that the near surface fluxes are fully within the sub-grid scale (SGS). When moving to LES scales, a portion of the fluxes are resolved, but the surface layer scheme underestimates the surface fluxes by only considering the SGS component in its calculation of the drag coefficient. Thus, in order to adhere to Monin-Obukhov Similarity Theory, the under-estimation of surface fluxes results in an erroneously large

drag coefficient. Additionally, this may also be due in part to the fact that the current model setup neglects the wind–wave relationship such as wave state, swell–wind alignment, etc. (Sullivan et al., 2008).

Lastly, performance is compared between setups on the mesoscale and microscale. We do this in order to answer the question of whether *we can assume that the best performing mesoscale setup will result in the best performance on the microscale?* While this analysis is subjective in how the "best" performers are determined for this single LLJ case, it represents a real-world

scenario faced by many scientists in the field. For this case, the best-performing mesoscale setup, the simulation with CMC SST data, ends up being the best-performing microscale setup as well. However, the second-best-performing mesoscale setup – and one that could potentially be chosen as the best setup depending on the metric of interest – becomes the worst-performing microscale setup overall. The ranking between best and worst mesoscale performing setups is not a one-to-one match with





the microscale ranking. This finding suggests that although we can try to set up our LES simulations to have the best chance
of success, the differences between the mesoscale and microscale numerical methods and model setup are large enough that
one of the best performers on the mesoscale may end up being the worst performer on the microscale. Conversely, one of the
worst mesoscale performers may produce the best results on the microscale. This finding is inherently tied to this case study
and not necessarily general. However, for cases in which the value of SST is consistent between the finest mesoscale domain
(domain 2) and the LES domains, we still see variation in performance across domains indicating that the differences between
turbulence closure (PBL scheme on the mesoscale and sub-grid turbulence parameterization on the LES) is enough to cause
"good" performance on the mesoscale to become "bad" performance on the LES.

This study also further elucidates the importance of further exploration into simulations within the *terra incognita*. Running
weather models at these resolutions violates most currently existing turbulence closure assumptions for both the mesoscale
and microscale. It has yet to be determined how best to deal with simulations at these scales whether that means skipping
over domains at this region (a large jump in parent grid ratio) or through the use of boundary layer parameterizations that
are more applicable at this scale (Shin and Hong, 2015; Kosovic et al., 2016; Kosović et al., 2020). Future work will focus
on the differences between mesoscale and microscale turbulence closure and surface layer parameterizations to determine the
underlying differences in numerics that consistently cause slower wind speeds near the surface in the LES domains and stronger
low-level shear.

*Code and data availability.* The NYSERDA floating lidar data were obtained through the OceanTech Services/DNV under contract to NY-
SERDA web portal: https://oswbuoysny.resourcepanorama.dnv.com/. Neither NYSERDA nor OceanTech Services/DNV have reviewed the
information contained herein and the opinions in this report do not necessarily reflect those of any of these parties.

**Appendix A**

The WRF Preprocessing System (WPS) is currently designed to compile with single precision. When utilizing small grid
sizes, $\Delta_x$, the calculation of variables COSALPHA and SINALPHA, which represent the components of the rotation angle
used in several map projections, are corrupted due to truncation issues when calculating the difference between latitude and
longitude over a cell. These variables, SINALPHA and COSALPHA, are then used within the Coriolis subroutine, gravity
wave damping, and in the psuedo-tower output (tslist) when rotating the winds to Earth coordinates. Additionally, many users
use these variables to rotate the model output winds to Earth coordinates in post processing. This issue has been found from
the earliest version of WPS that we could obtain and, thus, will impact any simulations that consider Coriolis and/or gravity
wave damping with a map projection other than Mercator. Additionally, any use of the tslist output winds or postprocessing
of winds to rotate to Earth coordinates will be impacted. This issue was unfortunately found in our model output after the
simulation suite was conducted, thus, tslist output was not used in this study and the issues associated with the COSALPHA
and SINALPHA calculations are embedded in our solution.





In order to determine if precision was the issue, we calculate COSALPHA and SINALPHA offline with double precision
(in our case, in Python) and then overwrite the variables in the geo_em.d0X.nc files. We tested this workaround by running
small test cases with two domains for 48 hours with and without overwriting COSALPHA and SINALPHA to determine the
impact on the wind speeds. While the calculation of COSALPHA and SINALPHA is not perfect (latitude and longitude were
not calculated based on a map projection), it serves as a simple test to determine the impact of the truncation issues. The $\Delta_x$ on

these domains was set to 2500 m and 500 m, respectively. The percent difference in the wind speeds grew with time and was
most significant when wind speeds were low due to small differences having a larger impact when wind speeds themselves are
small. The differences were also larger as $\Delta_x$ decreased in size. That said, the differences in wind speed never grew above 10%
and were on average closer to 1–2% throughout the simulation on both domains. This workaround does alleviate any issues in
the rotation of the winds. Thus, we are confident that the basic findings of our sensitivity study are not impacted by this WRF

issue. A permanent solution within WPS has been made and will be included in the next official release.



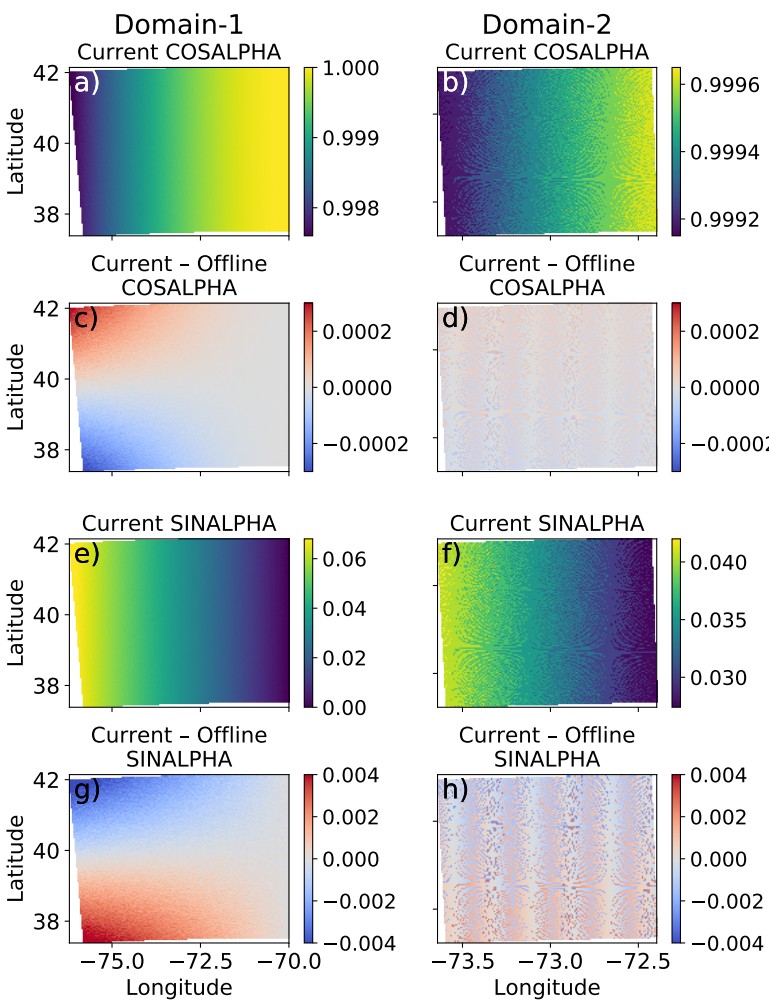

**Figure A1.** Contoured WRF-calculated COSALPHA on domains 1 and 2 (a and b, respectively), and the difference between the WRF calculation and offline calculation of COSALPHA using double precision for domains 1 and 2 (c and d, respectively). Contoured WRF-calculated SINALPHA on domains 1 and 2 (e and f, respectively), and the difference between the WRF calculation and offline calculation of SINALPHA using double precision for domains 1 and 2 (g and h, respectively).




**Figure A2.** Domain-averaged percent error between the simulations with the current WRF-calculated COSALPHA and SINALPHA and the offline calculated COSALPHA and SINALHPA along with the domain-averaged value for u-componend winds (a), v-component winds (b), and horizontal wind speed (c).

*Author contributions.* Conceptualization, all authors; methodology, all authors; software, PH and WL; validation, PH; formal analysis, PH; investigation, PH; data curation, PH and WL; writing—original draft preparation, PH; writing—review and editing, all authors; visualization, PH; funding acquisition, SEH. All authors have read and agreed to the published version of the manuscript.



*Competing interests.* No competing interests are present.

*Acknowledgements.* Funding was provided by the U.S. Department of Energy Office of Energy Efficiency and Renewable Energy Wind Energy Technologies Office. The views expressed in the article do not necessarily represent the views of the DOE or the U.S. Government. The U.S. Government retains and the publisher, by accepting the article for publication, acknowledges that the U.S. Government retains a nonexclusive, paid-up, irrevocable, worldwide license to publish or reproduce the published form of this work, or allow others to do so, for U.S. Government purposes. The National Science Foundation National Center for Atmospheric Research (NSF NCAR) was a subcontractor

to Pacific Northwest National Laboratory (PNNL), operated by the Battelle Memorial Institute, for the U.S. DOE under Contract No. DE-A06-76RLO 1830. Lawrence Livermore National Laboratory is operated by Lawrence Livermore National Security, for the U.S. DOE under Contract No. DE-AC52-07NA27344. NSF NCAR is a major facility sponsored by the National Science Foundation under Cooperative Agreement No. 1852977.

Portions of this work were performed under the auspices of the U.S. Department of Energy by Lawrence Livermore National Laboratory

under Contract DE-AC52-07NA27344. Neither NYSERDA nor OceanTech Services/DNV have reviewed the information contained herein and the opinions in this report do not necessarily reflect those of any of these parties.



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
