# Peer review of "Model sensitivity across scales: a case study of simulating an offshore low-level jet"

_Wind Energy Science, 2025_

## Referee Comment (RC1)

**Synopsis:**

The manuscript "Model sensitivity across scales: a case study of simulating an offshore low-level jet" by the authors P. Hawbecker, W. Lassman, T. W. Juliano, B. Kosovic and S. E. Haupt deals with the investigation of the sensitivity of mesoscale-to-microscale simulations of a specific case of an offshore low-level jet (LLJ) to varying sea surface temperature (SST) datasets. The authors aim to understand how the representation of physical characteristics of the observed LLJ changes between mesoscale and microscale simulations. Moreover, the authors investigate the question whether a better mesoscale setup leads also an improved result of the microscale simulation coupled to this mesoscale simulation.

The study shows for the specific case investigated that low-level shear and jet nose height are better represented in microscale simulations compared to mesoscale simulations. Specifically, low-level shear improves by reducing near-surface wind speeds and lowering the jet nose height to be closer to observations. The study highlights the challenges in predicting the mesoscale setup that will result in the best performance of the microscale model. While the best performing mesoscale setup (CMC SST data) ends up being the best performing microscale setup, the second-best performing mesoscale setup becomes the worst performing microscale setup. This suggests that the differences between mesoscale and microscale numerical methods and model setup are large enough that one of the best performers on the mesoscale may lead to one of the worst performances of the microscale model.

While the paper is overall well written and addresses with mesoscale-microscale coupling a hot topic in the field of wind energy research, my main concern refers to the question how generalizable the results presented by the authors actually are. I think it would strengthen the manuscript a lot if the authors could show that their conclusions from the single low-level jet case at one site can be transferred to other low-level jet cases at the same site and a next step also to low-level jet cases at another site. If no additional sites or measurements are included in the manuscript it should be mentioned more clearly that the reader should be careful to transfer the findings of this paper to other sites.

I have the following specific comments on the paper:

- 1. Line 64: Please change "shear over the rotor swept" to "shear over the rotor swept area"
- 2. Line 91: Please change "Initial and boundary conditions for model" to "Initial and boundary conditions for the model".
- 3. Line 91: I ask the authors to provide a motivation for using the MERRA-2 data for providing the initial and boundary conditions. E.g., it would be an important information if previous papers have shown that this reanalysis dataset performs best in the region under investigation by the authors.
- 4. Line 115: I ask the authors to add additional information on the correction of measurement data for tidal variation.
- 5. Figure 2: I found it slightly difficult to determine the height of the maximum wind speed from this plot (is it really always at about 120 m as stated by the authors in the text?). I think it would be helpful for the reader if the authors added markers showing the position of the maximum at each time to this figure.
- 6. Line 130: Is a resolution of 10 m for the simulation of a stably stratified marine atmospheric boundary layer that shows an LLJ sufficient for an LES? I think it would be good to slightly lower expectations already at this point.

- 7. Line 151: I ask the authors to please clarify whether the output that is produced every 10 minutes is instantaneous or time-averaged data.
- 8. Line 163: Please replace "closer" by "closure".
- 9. Line 178/179: The authors mention several times that it is computationally expensive to run the LES simulations. My suggestion is to provide information on the resources that were actually consumed for the simulations by the authors. How many core-h on what type of HPC infrastructure have been consumed?
- 10. Line 188/189: In recent years a number of criteria for detecting low-level jets have been suggested in literature. I think the authors should refer to these criteria.
- 11. Line 199: From my point of view the observation of a too strong shear in the microscale simulations of the authors was to be expected. The chosen resolution is too coarse to really be an LES of a stable atmospheric boundary layer. I expect that this leads to a too low turbulence in the model. Thus, the shear becomes too large.
- 12. Figure 5: Please add markers that provide information on the core height also in this figure.
- 13. Line 257/258: "The mesoscale domains befit from slightly under-predicting wind speeds below hub height and over-predicting wind speeds above" Doesn't this sentence contradict the result that the shear in the mesoscale model is low? (see e.g. figure 9) In Line 260 the authors state: "Analysing the time series of the ensemble mean of bias in low-level shear the mesoscale domains underpredict low-level shear while LES over-predict." This sounds contradictory to the sentence in line 257/258 to me.
- 14. Line 268: "It is interesting to note that the mesoscale domains produce larger negative values of SHFX than the LES domains, which indicates more stable conditions". Couldn't this be checked by checking the profile of potential temperature? Another possible explanation is that the eddy viscosity could be overestimated by the mesoscale model. This would fit to the wind profiles having less shear.

---

## Referee Comment (RC2)

Review of "Model sensitivity across scales: a case study of simulating an offshore low-level jet" by Patrick Hawbecker, William Lassman, Timothy W. Juliano, Branko Kosović, and Sue Ellen Haupt.

The manuscript provided by the authors addresses the application of different sea surface temperature (SST) data sources to model specific events across different modelling scales and validate them against floating lidar measurements. The objective of the presented study is to improve the understanding of the sensitivity of simulation frameworks across different scales on SST datasets and find the best performing dataset. Simulations are carried out in the Weather Research and Forecasting model (WRF) using five nested domains with increasing horizontal resolution, ranging from 6250 m to 10 m, thus covering the mesoscale, grey zone of turbulence and microscale. Specifically, the study deals with one offshore Low-Level Jet event observed by a floating lidar device off the coast of New Jersey in the United States comparing the measured physical features, such as vertical wind shear, jet core height and speed to the characteristics obtained by the seven simulation members driven by different SST datasets.

The authors found that the best-performing SST dataset throughout all domains is the CMC SST dataset. For some other datasets large discrepancies between performance in microscale and mesoscale domains is observed. The OSPO dataset for example performs second best in the mesoscale while showing the worst performance in microscale. The opposite is true for the OSTIA dataset, as it secures a third place in the microscale, while performing worst in the mesoscale. Further, the authors acknowledge, that these results are obtained from observations of one specific event at one location and also underlie the subjective weighting of different metrics to generate a ranking. Thus, the authors claim that while the process of determining good performing setups is well-suited, the overall ranking is not generalizable for other situations.

The article provides valuable inputs for the modelling community, but could benefit a lot from a broader discussion about the generalizability of it's findings, i.e. how the findings could be transferred for a broader audience facing similar challenges with different meteorological situations. Also, the mainly time-series based visualization of results sometimes is hard to follow and to interpret. Here, a more a more concise and systematically organized presentation would enhance their clarity and readability.

**General comments**

- 1. The introduction could benefit from short description about the formation of LLJs, i.e. why the temperature difference between sea surface and air plays a major role in their development.
- 2. When describing the considered LLJ event in Section 2.3, some meteorological information about the presented case, such as e.g. the wind direction, is missing. A more elaborate information about the event and

- why this specific case is of interest would be helpful in following the story of the manuscript.
- 3. Instead of relying on time series representations that much in the results, maybe a depiction of e.g. the correlation between the meteorological features would make it more easy to spot good or bad performing setups quickly. The way the results are presented right now is quite repetitive and differences between the different setups are hard to interpret.
- 4. In Section 4.4 you briefly touch the process of selecting the "best" performing setup to run the LES domains. While you recognize, that this process is subjective and depends on the region and event of interest, maybe you could elaborate a bit more on the process of how you chose which metrics are of importance and how you weigh them when reaching the conclusions. This more transparent approach would also be of great help in the Methods section to understand the reasoning behind the entire process.
- 5. In my opinion the manuscript would benefit of a separation of Results and Discussion. Right now, some discussion of the results is already performed in the Section 3, while some of it is also present in Section 5, making it sometimes hard to follow the storyline of your paper.

**Minor comments**

- 1. L. 64: Here an "area" is missing after "rotor swept"
- 2. L. 103: I do not fully understand why the highest resolved dataset (MUR) should not be smoother than lower granularity datasets such as OSTIA or GOES-16. In my mind the highest granularity dataset should provide the smoothest gradient. Could you elaborate why that is a contradiction?
- 3. L. 111 Here, it would be helpful to include the information, that the buoys are also used to gather the Temperature profiles and SST.
- 4. L. 163: If I am not mistaken it should read "LES turbulence *closure* techniques" not "*closer*"
- 5. L. 164: You mention the use of LES turbulence closure techniques is questionable for these resolutions. Could you maybe elaborate on why that is and why you still chose to use this parametrization over mesoscale PBL schemes which are "even more questionable"?
- 6. Figure 4: Consider changing the numbers to markers in the Figure, as this depiction looks rather confusing than helpful for me. Also, I'm not quite sure, what the spread here depicts. Is it just the difference between maximum and minimum value for the different SST datasets?
- 7. L. 193: "Wind speed maximum" instead of "maxima"

- 8. L. 200: What is here meant by "shifting" the data? Is this just a temporal shift? Regarding this point it would also be interesting whether grid or spectral nudging is used when driving the simulations. Could you please elaborate here?
- 9. L. 214: When referring to shear, are you talking about the bulk shear  $(\frac{z_{\rm up}-z_{\rm low}}{\Delta z})$  between lower tip and hub height, or the average shear across that region  $(\sum_i \frac{u(z_{i+1})-u(z_i)}{z_{i+1}-z_i})$ ?
- 10. L. 226: For consistency, please align the depiction of means. In Figure 8, they are shown as overlines in the Figure titles.
- 11. Figure 9: In all other figures, domains in the legend are not abbreviated. Consider aligning for consistency. The same is true for Figure A2.
- 12. Line 291: Are bias and RMSE calculated in reference to the Observations here?
- 13. Table 2: For brevity you could consider summarizing D01 and D02 as mesoscale, D03 as grey zone and D04 and D05 as LES domains. This would make the table more accessible as the results for both mesoscale and microscale domains, respectively, are the same anyhow.
- 14. L. 368: Is this a new research question? Consider already adding this part to your objective statement in the Introduction.

---

## Author Comment (AC1)

Original comments; Author responses

The manuscript provided by the authors addresses the application of different sea surface temperature (SST) data sources to model specific events across different modelling scales and validate them against floating lidar measurements. The objective of the presented study is to improve the understanding of the sensitivity of simulation frameworks across different scales on SST datasets and find the best performing dataset. Simulations are carried out in the Weather Research and Forecasting model (WRF) using five nested domains with increasing horizontal resolution, ranging from 6250 m to 10 m, thus covering the mesoscale,grey zone of turbulence and microscale. Specifically, the study deals with one offshore Low-Level Jet event observed by a floating lidar device off the coast of New Jersey in the United States comparing the measured physical features, such as vertical wind shear, jet core height and speed to the characteristics obtained by the seven simulation members driven by different SST datasets.

The authors found that the best-performing SST dataset throughout all domains is the CMC SST dataset. For some other datasets large discrepancies between performance in microscale and mesoscale domains is observed. The OSPO dataset for example performs second best in the mesoscale while showing the worst performance in microscale. The opposite is true for the OSTIA dataset, as it secures a third place in the microscale, while performing worst in the mesoscale. Further, the authors acknowledge, that these results are obtained from observations of one specific event at one location and also underlie the subjective weighting of different metrics to generate a ranking. Thus, the authors claim that while the process of determining good performing setups is well-suited, the overall ranking is not generalizable for other situations. The article provides valuable inputs for the modelling community, but could benefit a lot from a broader discussion about the generalizability of it's findings, i.e. how the findings could be transferred for a broader audience facing similar challenges with different meteorological situations. Also, the mainly time-series based visualization of results sometimes is hard to follow and to interpret. Here, a more a more concise and systematically organized presentation would enhance their clarity and readability.

We would like to thank the reviewer for their summary and suggestions for improvements to the manuscript. The suggestions for improvements above are itemized in the general comments and are addressed below with the exception of the generalizability of the study. We appreciate the request to address the generalizability of these findings and have added the following text to the manuscript:
Line 69 of original manuscript:
*We note that the study considers a single case study for a specific topic; thus, it is unclear whether the findings generalize to other cases and atmospheric phenomena. However, we explore fundamental differences in mesoscale and microscale simulation techniques that are generally applicable for other atmospheric studies.*
Line 375 of original manuscript:
*These fundamental differences emphasize that studies should generally use caution when assuming that mesoscale sensitivities will directly translate to the microscale when simulating atmospheric phenomena (such as low-level jets) that have known dependencies on model grid spacing and turbulence closure techniques.*

Original comments; Author responses

*Line 383 of the original manuscript:*
*Additionally, similar studies for additional cases, different atmospheric phenomena, and different parametric sensitivities will be required in future studies to determine when and where mesoscale sensitivity can directly translate to microscale sensitivity.*

General Comments

1.  The introduction could benefit from short description about the formation of LLJs, i.e. why the temperature difference between sea surface and air plays a major role in their development.
    We appreciate the reviewer's suggestion and have added the following text to the manuscript:
    Line 47 of the original manuscript:
    Offshore low-level jets can form when relatively warmer air advects over colder water temperatures generating stable conditions and frictional decoupling of the winds aloft (see De Jong et al., (2024) for more information on offshore low-level jet formation mechanisms in this region).

2.  When describing the considered LLJ event in Section 2.3, some meteorological information about the presented case, such as e.g. the wind direction, is missing. A more elaborate information about the event and why this specific case is of interest would be helpful in following the story of the manuscript.
    This is a good suggestion and the following text has been added to the manuscript:
    Line 114 of the original manuscript:
    The dominant wind direction in this case was from the South indicating the possibility of warm air advection as is commonly seen in the region (De Jong et al., 2024).
    Line 120:
    This case was selected due to the clear and consistent LLJ signal and the apparent dependence on the air-sea temperature gradient.

3.  Instead of relying on time series representations that much in the results, maybe a depiction of e.g. the correlation between the meteorological features would make it more easy to spot good or bad performing setups quickly. The way the results are presented right now is quite repetitive and differences between the different setups are hard to interpret.
    We appreciate the suggestion of the reviewer but do not agree that the analysis is redundant, and in fact, by comparing many variables over time, one can compare against variables/Figures as is done in the manuscript. It is discouraging to hear that the reviewer finds the analysis repetitive but we do not feel it is helpful to change the presentation of the results for the sake of variety.
    We do not believe that the additional analysis of correlation would necessarily reveal better performing models and do not think it is pertinent to this study. In this study we considered bias and error; other studies may value correlation more, but that is highly dependent on the goals of the study.

4.  In Section 4.4 you briefly touch the process of selecting the "best" performing setup to run the LES domains. While you recognize, that this process is subjective and depends on the region and event of interest, maybe you could elaborate a bit more on the

process of how you chose which metrics are of importance and how you weigh them when reaching the conclusions. This more transparent approach would also be of great help in the Methods section to understand the reasoning behind the entire process.

We thank the reviewer for recognizing how this section relies on subjective interpretation and have added the following text to clarify our ranking:

Line 307 of the original manuscript:

*Ranking the performance is highly dependent on the specific feature being studied. For this study, we consider performance of the dynamic variables above (low-level shear, hub-height wind speed, and REWS) and forcing conditions (2-m Temperature, SST, and ΔT) to rank the mesoscale setup performance. Considering these variables and weighing them equally, we rank the mesoscale SST dataset performance from best to worst as follows: CMC, OSPO, GOES-16, Default SST, NAVO, MUR, OSTIA (Table 2).*

5. In my opinion the manuscript would benefit of a separation of Results and Discussion. Right now, some discussion of the results is already performed in the Section 3, while some of it is also present in Section 5, making it sometimes hard to follow the storyline of your paper

We appreciate the comment from the reviewer on how to make the storyline more coherent. We do not see where in Section 3 (Model Setup) that results are discussed so unless specific text is noted, we cannot address this concern. Section 5 contains the discussion of results and has been renamed to "Summary and Discussion" rather than "Summary and Conclusions." There are areas in Section 4 (Results) in which the results are contextualized to provide readers with justification for why a result is worth noting. While we could move that text to Section 5, we believe that the results section would be more difficult to follow. The beginning of the results section lays out the organization of the results and Section 5 brings the results into context. We welcome specific comments as to where the reviewer thinks the storyline could be improved.

**Minor Comments:**

1. L. 64: Here an "area" is missing after "rotor swept"

   Thank you for finding this error; it has been corrected in the revised manuscript.

2. L. 103: I do not fully understand why the highest resolved dataset (MUR) should not be smoother than lower granularity datasets such as OSTIA or GOES-16. In my mind the highest granularity dataset should provide the smoothest gradient. Could you elaborate why that is a contradiction?

   We thank the reviewer for the question but would like to suggest that the highest granularity dataset does not necessarily imply the sharpest gradients. We agree that the MUR dataset does appear to resolve less energy at small scales than other datasets, but describing how each of these products is generated is beyond the scope of this paper.

3. L. 111 Here, it would be helpful to include the information, that the buoys are also used to gather the Temperature profiles and SST.

   We have adjusted the sentence on line 107 by adding the following text to the manuscript to clarify this:

   Line 107 of the original manuscript:

   *These buoys, named E05 and E06 (open and filled X in Figure1, respectively), contain*

*vertically scanning lidars, ocean and wave sensors, and a small meteorological mast recording atmospheric variables such as temperature and pressure.*

4. L. 163: If I am not mistaken it should read "LES turbulence closure techniques" not "closer"

Thank you for finding this error; it has been corrected in the revised manuscript.

5. L. 164: You mention the use of LES turbulence closure techniques is questionable for these resolutions. Could you maybe elaborate on why that is and why you still chose to use this parametrization over mesoscale PBL schemes which are "even more questionable"?

We thank the reviewer for their question and have added the following text to the manuscript:

Line 157 of the original manuscript:

*LES turbulence closure techniques are used in this study within this domain although its appropriateness is questionable due to the fact that the largest energy-containing eddies are not fully resolved with this grid spacing. The other option is to use a planetary boundary layer scheme at this resolution, but the assumptions of horizontal homogeneity and that all energy-containing eddies are unresolved renders the applicability of such parameterizations at 250 m grid spacing are yet more questionable.*

6. Figure 4: Consider changing the numbers to markers in the Figure, as this depiction looks rather confusing than helpful for me. Also, I'm not quite sure, what the spread here depicts. Is it just the difference between maximum and minimum value for the different SST datasets?

We thank the reviewer for the suggestion. We have tried using symbols for this figure but the same issue crops up where all or most of the symbols are on top of one another and in the end, the figure is not more clear. Considering that, we found the numbers more useful as there is less question about what "1" represents (domain-1); whereas if a circle were used, it is not as obvious and readers are left mapping the caption to the figure. Instead, the figure highlights that other than domain 1, the majority of the domains have similar representations of sea surface temperature in the domains. The spread quantifies exactly how different the average SST is between domain 1 and 5 as noted in the caption of Figure 4.

7. L. 193: "Wind speed maximum" instead of "maxima"

Thank you for this suggestion. The sentence has been reworded to be, "During the period of interest, maximum wind speeds are observed between 80 and 150 m (Figure 5a)

8. L. 200: What is here meant by "shifting" the data? Is this just a temporal shift? Regarding this point it would also be interesting whether grid or spectral nudging is used when driving the simulations. Could you please elaborate here?

We see how this is unclear. The sentence has been adjusted to read, "When shifting the observations in time to better match with simulated results…"

9. L. 214: When referring to shear, are you talking about the bulk shear between lower tip and hub height, or the average shear across that region?

On line 185-186 of the original manuscript we define what we mean by low-level shear but have added the term "bulk shear" to indicate that it is indeed the former. The

sentence now reads, "We define low-level shear as the bulk shear between the bottom of the rotor swept area (38~m) and hub-height"

10. L. 226: For consistency, please align the depiction of means. In Figure 8, they are shown as overlines in the Figure titles.
We thank the reviewer for the suggestion but are unsure as to what needs to be adjusted. Line 220 notes that angle brackets represent ensemble average, while Line 229 explains that the overbar represents time average. We do not see an instance in which this is misused and would appreciate it if the reviewer clarified where this is done so we can change it.

11. Figure 9: In all other figures, domains in the legend are not abbreviated. Consider aligning for consistency. The same is true for Figure A2.
We thank the reviewer for catching this inconsistency. The figures have been updated in the revised manuscript.

12. Line 291: Are bias and RMSE calculated in reference to the Observations here?
Yes. To clarify this, the sentence now reads, "Considering root mean square error (RMSE) and bias with respect to observations for each setup on the mesoscale and microscale domains…"

13. Table 2: For brevity you could consider summarizing D01 and D02 as mesoscale, D03 as grey zone and D04 and D05 as LES domains. This would make the table more accessible as the results for both mesoscale and microscale domains, respectively, are the same anyhow.
We thank the reviewer for the suggestion but disagree that mesoscale and microscale columns should be joined because the performance rankings are the same on both mesoscale domains and on both microscale domains. This explicitly shows that the performance rankings do not change much between mesoscale-mesoscale and microscale-microscale. We believe that combining these columns also would not be consistent with the rest of the analysis performed in which all domains are shown separately.

14. L. 368: Is this a new research question? Consider already adding this part to your objective statement in the Introduction.
It is unclear what statement the reviewer is referring to with this comment. We are assuming it is about the sentence reading, "This finding suggests that although we can try to set up our LES simulations to have the best chance of success, the differences between the mesoscale and microscale numerical methods and model setup are large enough that one of the best performers on the mesoscale may end up being the worst performer on the microscale."
We believe that this is addressing one of the central research questions of the study of, "can we expect the microscale simulation driven by the best performing mesoscale setup to produce the best microscale result?"

---

## Author Comment (AC2)

Original comments; Author responses

The manuscript "Model sensitivity across scales: a case study of simulating an offshore low-level jet" by the authors P. Hawbecker, W. Lassman, T. W. Juliano, B. Kosovic and S. E. Haupt deals with the investigation of the sensitivity of mesoscale-to-microscale simulations of a specific case of an offshore low-level jet (LLJ) to varying sea surface temperature (SST) datasets. The authors aim to understand how the representation of physical characteristics of the observed LLJ changes between mesoscale and microscale simulations. Moreover, the authors investigate the question whether a better mesoscale setup leads also an improved result of the microscale simulation coupled to this mesoscale simulation.
The study shows for the specific case investigated that low-level shear and jet nose height are better represented in microscale simulations compared to mesoscale simulations. Specifically, low-level shear improves by reducing near-surface wind speeds and lowering the jet nose height to be closer to observations. The study highlights the challenges in predicting the mesoscale setup that will result in the best performance of the microscale model. While the best performing mesoscale setup (CMC SST data) ends up being the best performing microscale setup, the second-best performing mesoscale setup becomes the worst performing microscale setup. This suggests that the differences between mesoscale and microscale numerical methods and model setup are large enough that one of the best performers on the mesoscale may lead to one of the worst performances of the microscale model.

While the paper is overall well written and addresses with mesoscale-microscale coupling a hot topic in the field of wind energy research, my main concern refers to the question how generalizable the results presented by the authors actually are. I think it would strengthen the manuscript a lot if the authors could show that their conclusions from the single low-level jet case at one site can be transferred to other low-level jet cases at the same site and a next step also to low-level jet cases at another site. If no additional sites or measurements are included in the manuscript it should be mentioned more clearly that the reader should be careful to transfer the findings of this paper to other sites.

We would like to thank the reviewer for their comments and suggestions for this paper. We appreciate the need to further address the generalizability of these findings and have done so through adding the following text to the manuscript:
Line 69 of original manuscript:
*We note that the study considers a single case study for a specific topic; thus, it is unclear whether the resulting findings generalize to other cases and atmospheric phenomena. However, we explore fundamental differences in mesoscale and microscale simulation techniques that are generally applicable for other atmospheric studies.*
Line 375 of original manuscript:
*These fundamental differences emphasize that studies should generally use caution when assuming that mesoscale sensitivities will directly translate to the microscale when simulating atmospheric phenomena (such as low-level jets) that have known dependencies on model grid spacing and turbulence closure techniques.*

With that said, it is not feasible at this time to perform this analysis for another case. We have included this idea in the summary as future work. The following text has been added to the manuscript:
Line 383 of the original manuscript:
*Additionally, similar studies for additional cases, different atmospheric phenomena, and different parametric sensitivities will be required in future studies to determine when and where mesoscale sensitivity can directly translate to microscale sensitivity.*

We agree that it is possible that these findings are not completely generalizable to all situations. However, this is meant as a caution to people who are making the assumption that their mesoscale sensitivity study will in some way ensure high levels of performance on the microscale. It is possible, and is shown here, that depending on the metrics of interest and what is deemed important for a study, the best mesoscale simulation setup may not lead to the best microscale setup due to the differences in mesoscale and microscale numerics and turbulence closure.

I have the following specific comments on the paper:
1. Line 64: Please change "shear over the rotor swept" to "shear over the rotor swept area"
   Thank you for catching this mistake. It has been fixed in the revised manuscript.
2. Line 91: Please change "Initial and boundary conditions for model" to "Initial and boundary conditions for the model"
   Thank you for catching this mistake. It has been fixed in the revised manuscript.
3. Line 91: I ask the authors to provide a motivation for using the MERRA-2 data for providing the initial and boundary conditions. E.g., it would be an important information if previous papers have shown that this reanalysis dataset performs best in the region under investigation by the authors.
   Thank you for mentioning this. The following text has been added to the manuscript in original line 138 to provide reasoning to the choice of MERRA-2 as the boundary condition:
   *The European Centre for Medium-Range Weather Forecasts (ECMWF) v5 reanalysis (Hersbach et al., 2020, ERA5) dataset was also tested and performance with MERRA-2 was found to be slightly better for the specific case day considered here (not shown).*
4. Line 115: I ask the authors to add additional information on the correction of measurement data for tidal variation.
   We'd like to thank the reviewer for pointing this out. This statement is erroneous and needs to be fixed. The lidar data are not corrected for waves/tides as they are averaged over 10 minutes. When averaging over 10 minutes, it has been shown that correcting for waves is unnecessary for these purposes (see Figure 14c in the reference below).
   Krishnamurthy, R., García Medina, G., Gaudet, B., Gustafson Jr, W. I., Kassianov, E. I., Liu, J., ... & Mahon, A. M. (2023). Year-long buoy-based observations of the air–sea transition zone off the US west coast. Earth System Science Data Discussions, 2023, 1-53.
   The original sentence in question has been removed and the following text has been added to the manuscript on Line 115 of the original manuscript:
   *The data are available at 10-minute averaged output and are not corrected for wave or tidal*

*variation. This correction has been shown to have a negligible effect in offshore floating lidar measurements when averaged at 10 minute timescales (Krishnamurthy et al. 2023).*

5. Figure 2: I found it slightly difficult to determine the height of the maximum wind speed from this plot (is it really always at about 120 m as stated by the authors in the text?). I think it would be helpful for the reader if the authors added markers showing the position of the maximum at each time to this figure.

This is a great idea and markers have been added to the figure to denote jet height. Additional text to explain this has also been added to the caption.

6. Line 130: Is a resolution of 10 m for the simulation of a stably stratified marine atmospheric boundary layer that shows an LLJ sufficient for an LES? I think it would be good to slightly lower expectations already at this point.

This is a fair point and we agree that expectations should be properly set for the ability of the LES to fully resolve turbulence in a stable boundary layer with 10 m grid spacing. The following text has been added to the manuscript on Line 124 of the original manuscript:
*Note that even with a $\Delta x$ of 10 m, we are likely not fully resolving the inertial subrange of the stable boundary layer and associated low-level jet (Beare and Macvean, 2004; Beare et al., 2006). Simulations at higher resolutions (down to sub-meter grid spacing) have not shown clear convergence in simulating the very stable boundary layer (Sullivan et al., 2016) and a small ensemble of cases at these resolutions is out of the scope of this current project. We do not anticipate the overall findings from this study to be impacted by the use of 10 m horizontal grid spacing as all simulations are equally impacted.*

7. Line 151: I ask the authors to please clarify whether the output that is produced every 10 minutes is instantaneous or time-averaged data.

Thank you for bringing up this oversight. Yes, the model data is also averaged in time. The following line has been added to line 165 in the original text:
*Thus, the simulation datasets are also averaged at 10-minute intervals for comparison with observations.*

8. Line 163: Please replace "closer" by "closure"

Thank you for catching this spelling mistake. It has been fixed in the revised manuscript.

9. Line 178/179: The authors mention several times that it is computationally expensive to run the LES simulations. My suggestion is to provide information on the resources that were actually consumed for the simulations by the authors. How many core-h on what type of HPC infrastructure have been consumed?

This is a very good point. The following has been added to the manuscript to provide evidence of the computational expense:
*The simulations were run on the NSF National Center for Atmospheric Research (NCAR) Cheyenne supercomputer. To provide evidence of the computational expense of LES for real-data cases, the following information is provided. Each LES run required 1,296~cores running for between 11 and 12 hours of wall-clock time in order to produce 10~minutes of simulation. Thus, to run for the full 6 hours, 35 restarts were required to fit within the Cheyenne 12-hour job limit. In total, this results in between 513,000~to~560,000 core hours per LES simulation -- 3.6-3.9~million~core hours in total.*

10. Line 188/189: In recent years a number of criteria for detecting low-level jets have been suggested in literature. I think the authors should refer to these criteria.

We appreciate the reviewer's suggestion and have added the following text to the manuscript:

On line 114: "While there exist many techniques to detect low-level jets from wind profiles (see, for example, Piety (2005) and Hallgren et al. (2023)), the observed low-level jet was detected using the technique described in Debnath et al. (2021)."

On line 188: "The height of maximum wind speed is used to define the low-level jet height."

11. Line 199: From my point of view the observation of a too strong shear in the microscale simulations of the authors was to be expected. The chosen resolution is too coarse to really be an LES of a stable atmospheric boundary layer. I expect that this leads to a too low turbulence in the model. Thus, the shear becomes too large.

This is a good perspective to mention and we will include this as a possibility for the overestimation of low-level shear within the discussion. The portion of the paragraph starting on line 352 of the original manuscript now reads:

*This finding leads us to believe that one or multiple of the following scenarios are occurring:*
*– The mesoscale MYNN 2.5 PBL parameterization may overly mix the stable boundary layer*
*– The 1.5 order TKE sub-grid turbulence scheme on the LES domains misrepresents surface drag over the ocean*
*– The grid spacing on the LES domains is too large to resolve turbulence within the stable boundary layer, resulting in an overproduction of shear*

12. Figure 5: Please add markers that provide information on the core height also in this figure.

This is a great idea and markers have been added to the figure to denote jet height. Additional text to explain this has also been added to the caption.

13. Line 257/258: "The mesoscale domains befit from slightly under-predicting wind speeds below hub height and over-predicting wind speeds above" Doesn't this sentence contradict the result that the shear in the mesoscale model is low? (see e.g. figure 9) In Line 260 the authors state: "Analysing the time series of the ensemble mean of bias in low-level shear the mesoscale domains underpredict low-level shear while LES over-predict." This sounds contradictory to the sentence in line 257/258 to me.

This is admittedly a confusing sentence and the word "slightly" was meant to apply to both the underprediction and overprediction of wind speeds below and above the jet nose, respectively. The sentence has been reworded to be:

*While agreement between the observations and simulations is decent below 100~m, the mesoscale domains do not overpredict the wind speeds as much as the LES domains do above the jet height (Figure 8d), which reduces error in REWS.*

14. Line 268: "It is interesting to note that the mesoscale domains produce larger negative values of SHFX than the LES domains, which indicates more stable conditions" Couldn't this be checked by checking the profile of potential temperature? Another possible explanation is that the eddy viscosity could be overestimated by the mesoscale model. This would fit to the wind profiles having less shear.

In terms of the stability, we appreciate the suggestion to check the potential temperature profiles and we had done so in our analysis, but did not include the plots in the manuscript. Although it may be helpful to add to the manuscript,we do not believe that adding an additional figure is necessary to make this point. The following sentence has been added in Line 262 of the original manuscript:

Original comments; Author responses

*This is reinforced when checking the potential temperature profiles in which the lapse rate near the surface for the mesoscale domains is stronger than that of the LES domains (not shown).*
With respect to eddy viscosity, this is indeed a possibility and we suggest this in the conclusions without explicitly mentioning eddy viscosity (Line 353 of the original manuscript). We agree that it is important to mention the mechanism that may be the cause of the over-mixing. The following sentence has been added to Line 353 of the original manuscript:
*[T]he mesoscale MYNN 2.5 PBL parameterization may overly mix the stable boundary layer potentially as a result of over-predicting eddy viscosity…*

---

## Author Comment (AC3)

Original comments; Author responses

I have a minor comment regarding your response to Reviewer 2's general comment 3. I agree with the reviewer that repeatedly using time-series plots can become redundant; however, this does not mean that these plots should be removed. Rather, they should be complemented with additional quantitative analyses. Reviewer 2's comment is intended as constructive feedback, not as discouragement. While time series are indeed a useful way to compare different variables qualitatively, quantitative information is equally important. The manuscript should not require the reader to perform the analysis themselves. In this sense, Reviewer 2's suggestion regarding correlation analysis is valuable, as it points toward a quantitative complement to the time-series figures.

We would like to thank the editor for this clarification and have moved the figures that show RMSE and bias to the location of the timeseries plots along with adding relevant text connecting the time series to RMSE and bias.

We have analyzed correlation as suggested by the reviewer and found that the p-values for these calculations are very high and should not be considered statistically significant:

**Table 2.** Two-sided p-value for the Pearson-r correlation coefficients for low-level wind shear, hub-height wind speed, and rotor equivalent wind speed (REWS) for each model setup and each domain.

| | Low-Level Shear | | | | | Hub-Height Wind Speed | | | | | REWS | | | | |
|---|---|---|---|---|---|---|---|---|---|---|---|---|---|---|---|
| | d01 | d02 | d03 | d04 | d05 | d01 | d02 | d03 | d04 | d05 | d01 | d02 | d03 | d04 | d05 |
| Default | 0.997 | 0.331 | 0.273 | 0.302 | 0.264 | 0.443 | 0.840 | 0.524 | 0.445 | 0.347 | 0.637 | 0.993 | 0.994 | 0.847 | 0.770 |
| CMC | 0.551 | 0.468 | 0.142 | 0.152 | 0.163 | 0.635 | 0.478 | 0.373 | 0.508 | 0.489 | 0.707 | 0.969 | 0.890 | 0.855 | 0.820 |
| OSPO | 0.825 | 0.667 | 0.757 | 0.519 | 0.479 | 0.463 | 0.421 | 0.013 | 0.057 | 0.166 | 0.588 | 0.698 | 0.412 | 0.317 | 0.324 |
| MUR | 0.768 | 0.267 | 0.414 | 0.373 | 0.338 | 0.933 | 0.929 | 0.886 | 0.918 | 0.849 | 0.675 | 0.925 | 0.930 | 0.791 | 0.710 |
| NAVO | 0.655 | 0.142 | 0.132 | 0.225 | 0.253 | 0.600 | 0.766 | 0.473 | 0.680 | 0.540 | 0.364 | 0.981 | 0.888 | 0.748 | 0.667 |
| OSTIA | 0.880 | 0.535 | 0.499 | 0.487 | 0.442 | 0.192 | 0.259 | 0.717 | 0.555 | 0.679 | 0.712 | 0.839 | 0.972 | 0.771 | 0.724 |
| GOES16 | 0.594 | 0.167 | 0.191 | 0.294 | 0.364 | 0.753 | 0.954 | 0.368 | 0.727 | 0.830 | 0.411 | 0.851 | 0.886 | 0.877 | 0.896 |

Common thresholds for the p-value to state the correlations are significant are from 0.01 to 0.05. None of the correlations here meet that threshold. While we understand the interest in correlation, the amount of data is insufficient to allow for a reasonable representation of how the observations and simulations are linearly related. We hope the reorganization of figures and added context are sufficient to break up the time series figures, highlight the quantitative analysis that was performed, and overall help the flow of results.

Concerning the reviewer's remark about including a "Discussion" section, I would like to kindly remind you that such a section is not meant simply to reiterate or discuss the results, but rather to provide a critical reflection on the model, the numerical setup, its limitations, etc.... In other words, the Discussion section aims to look at the "big picture." I recognize that the term "Discussion" can sometimes be confusing, so I hope this clarification is helpful.

We thank the editor for this clarification and have reorganized the paper to have an entirely separate discussion section following the summary that highlights the big picture as well as limitations and future work.

---

## Referee Report (RR1)

The authors revised their manuscript and implemented helpful changes to improve the manuscripts quality. Below some further comments on the revised version of the manuscript are collected. To proceed I would recommend the paper to be accepted, subject to minor revisions.

1. L.9: There seems to be a typo: "scehme" instead of "scheme"

2. L.131: Thank you for clarifying the used LLJ detection algorithm. In my opinion, it would be helpful to the reader to also directly describe the used definition, since it — as described in Debnath et al. (2021) — uses a combination of shear and fall-off criteria. This might be useful information for the reader down the line, when you analyse the low-level shear between different simulation set-ups.

3. Similarly, it would be helpful, to see what values for shear and fall-off were detected in this specific event (e.g. in L. 216ff)

4. Fig. 7/ Fig.8: From the time series data in Fig. 7 and the profiles in Fig. 8, it is seen that the low-level shear, as well as the fall-off is considerably smaller for the mesoscale domains. Could you elaborate on whether the LLJ definition you applied, detects the LLJ throughout all domains and all different set-ups.

5. L.285/Fig. 9: For both hub height wind speed and REWS, EMEs larger than $1\,\mathrm{ms}^{-1}$ occur at times. Given that you already calculated the REWS, would it be possible to elaborate on how these differences in wind speed translate to differences in possible power production, as power changes with the cube of the velocity. I see, that your specific case shows wind speeds that are probably above rated wind speed for the turbine sizes you assume. This actually makes it a two-part comment: a) How do the EMEs and Spreads translate to lower wind speeds and b) how large is their effect on an exemplary turbine's power production?

---

## Author Response (AR2)

Original comments; Author responses

Below are point-by-point responses to the editor and both reviewers.

**Editor Comments**
The two reviewers noted substantial improvement of the manuscript and recommend publication after minor revision. Please read their comments carefully and address them adequately. I will perform the final check after your revision.

In addition, based on my own reading, I have the following observations:

1. Methods content should be moved out of the Results section
   Some material currently in Section 4.3 (in particular, Equations 1–3 and the description of ensemble metrics) belongs in Section 2 ("Methods"). The Results section should present only the outcomes and their interpretation, not methodological explanations. Moving these elements to Section 2 will significantly improve clarity. (By analogy: the recipe is given before the cake is baked, not after.)
   We understand this correction and the equations and descriptions have been moved to the methods section.

2. Missing units in Figures 9, 10, and 11
   Figures 9, 10, and 11 appear to be missing units on the y-axes (temperature, wind speed, shear, etc.). Please revise the axis labels accordingly. The font size of the figures can be slightly reduced if needed to accommodate units.
   We thank the editor for catching this oversight. Units have been added to all figures with missing units.

3. Clarification needed for Figure A1
   In Figure A1, the right-hand panels (domain 2) show aliasing artefacts that could mislead readers. In addition, the y-axes of these right-hand panels are not labelled, implying they are identical to the left-hand panels (domain 1). Please confirm whether that is correct and ensure the figure is unambiguous. I also note that "domain 1" and "domain 2" terminology has already been used in Figure 3. Are these the same domain? Consistency and clarity here are important.
   We recognize now that the Appendix was poorly describing the problem and have modified the text appropriately. The Appendix addresses the artifacts within the figures and the workarounds we made to approach it correctly. They emphasize that if using WRF at high resolution and WPS is used in single precision, you will have errors within your simulations. This issue was reported to the WRF/WPS developers and corrections have been added to the official release of WPS version 4.5.
   The labels in the y-axis have been added and the names have been changed to Domains 1A and 2A to further emphasize that the domains are different from the main sections of the paper.

Original comments; Author responses

4. Possible relocation of Figure 12
   Figure 12 quantifies air–sea temperature differences across SST datasets and domains. Because this information characterizes the inputs before the simulations are run, it may be better placed in Section 2 or Section 3 rather than in the Results section.
   This figure includes both inputs (SST) and outputs from the model (2-m T and delta T), thus it seems best to leave it after the model calculations are explained. We have modified the figure caption to clarify this point.
   "Same as Figure 7 but for 2-m temperature, SST, and $\Delta T$. While SST is defined from the input SST datasets, both 2~m temperature and the resulting $\Delta T$ are predicted within the model."

5. Clarification and strengthening of the quantitative comparison (reviewers' main remaining concern)
   Reviewer 1 notes that the manuscript would benefit from a clearer and more complete presentation of the quantitative results, particularly those related to the correlation analysis. Your response mentions high p-values, but does not specify which variables were correlated with which others. As a result, it is difficult to interpret Table 2 or understand the attempted correlation analysis.
   We see that we had neglected to clarify what was being compared in the correlation analysis. As with the other metrics in the study, the p-values from the correlations shown in Table 2 compares model vs. observations. Variables considered are noted in the top of Table 2: low-level shear, hub-height wind speed, and REWS.

   There may have been a misunderstanding around the word "correlation." Reviewer 2 was referring to the correlation between meteorological features. Specifically, they suggested a concise quantitative way to see how environmental inputs relate to model performance. Examples would include:
   Correlation (or simply quantitative metric) that shows the link between $\Delta T$ (air–sea temperature difference) and low-level shear error, between SST bias and hub-height wind speed bias, etc... Such relationships can be presented descriptively (e.g., correlation coefficients, scatter plots, etc...), without relying on statistical significance tests. The aim is interpretability: helping the reader see which setups tend to perform better and why.
   Unfortunately, analyzing the relationships of the suggested variable combinations ($\Delta T$ vs low-level shear error; SST-bias vs. hub-height wind speed, etc.) do not make things more interpretable for the limited number of hours modeled. Considering $\Delta T$ vs low-level shear difference (shown below), we see no notable relationship and the significance of the resulting correlation coefficients is above 0.1 for 75% of the datasets.

[Figure]

For the correlations that are of significance (p-value < 0.1) between ΔT vs low-level shear difference (table below), we see that they are either roughly -0.4 or +0.4 (and average to nearly zero). Unfortunately, this does not add additional insight to the study.

| Setup-domain | Corr. | Sig. |
|---|---|---|
| Default-d01 | -0.402 | 0.046 |
| CMC-d01 | -0.423 | 0.035 |
| CMC-d03 | 0.446 | 0.026 |
| CMC-d04 | 0.438 | 0.029 |
| CMC-d05 | 0.446 | 0.026 |
| NAVO-d01 | -0.456 | 0.022 |
| OSTIA-d01 | -0.425 | 0.034 |
| GOES16-d01 | -0.340 | 0.096 |
| GOES16-d03 | 0.406 | 0.044 |

SST-bias is effectively a constant value, so correlation between that and any other variable is undefined.

We have analyzed the data in a quantitative manner within this study and have not

included correlations because there simply is not enough data to draw conclusions on these relationships. We strongly believe that correlation between any variables without analysis of the statistical significance does not belong in a scientific journal. The statistical significance determines whether the results are meaningful.

More generally, the paper would be clearer if its structure more explicitly reflected the two-step logic of the study: (1) How the choice of SST dataset and domain affects the modelled meteorological state (this is well presented), and (2) How these differences translate into agreement or disagreement with the lidar observations (this is the part that may have been insufficiently quantified). I think that addressing even better the quantitative link between (1) and (2) would fully address the remaining reviewer concerns.

We agree with the reviewer that step (1) is well presented in the paper, but believe that step (2) is also well presented. The lidar observations are included in every metric shown with the exception of surface sensible heat flux in Figure 12c. The domain effects are highlighted in several areas in which the difference between the boundary layer parameterization and subgrid turbulence model drastically change the simulated LLJ characteristics. We do not heavily focus on the impacts of SST directly as it is not within the scope of this paper. To clarify the paper's intentions, we have adjusted the introduction to include the following paragraph:

"*In this study, model sensitivity of an offshore low-level jet (LLJ) to sea-surface temperature (SST) is analyzed across both the mesoscale and microscale.*

*The goal is to assess, on both the mesoscale and microscale, the sensitivity of LLJ characteristics to SST and model performance when compared to observations in order to determine whether the assumptions above are indeed valid.*"

**Reviewer 1**

I would like to thank the authors for carefully revising their manuscript. Once the point raised by the second reviewer and the editor on the extension of the presentation of quantitative results has been properly addressed by the authors it can be accepted for publication. I detected only one typo in the revised manuscript. In line 351 "weighing" should be replaced by "weighting".

We thank the reviewer for their comment and in catching this spelling error. It has been replaced in the manuscript.

**Reviewer 2**

The authors revised their manuscript and implemented helpful changes to improve the manuscripts quality. Below some further comments on the revised version of the manuscript are collected. To proceed I would recommend the paper to be accepted, subject to minor

Original comments; Author responses

revisions.
We thank the reviewer for their comments and suggestions.

1. L.9: There seems to be a typo: "scehme" instead of "scheme"
   We thank the reviewer for catching this error and have updated it in the manuscript.

2. L.131: Thank you for clarifying the used LLJ detection algorithm. In my opinion, it would be helpful to the reader to also directly describe the used definition, since it — as described in Debnath et al. (2021) — uses a combination of shear and fall-off criteria. This might be useful information for the reader down the line, when you analyse the low-level shear between different simulation set-ups.
   We have added a short description of the method to the manuscript but want to note that while the observed LLJ was detected using the Debnath et al. (2021) definition, the simulated LLJs are defined strictly by the height of maximum wind speed. Once detected in observations, we no longer use that algorithm.
   Added text: *"This algorithm detects a low-level jet based on meeting three criteria: (1) the maximum wind speed is not at the first or last lidar level, (2) the level of shear between the lowest lidar level and maximum wind speed is above $0.035~s$^{-1}$, and (3) the wind speed drop off between the maximum wind speed and top lidar measurement is greater than $1.5~m~s$^{-1}$ and the drop-off is more than 10\% of the maximum wind speed."*
   Similarly, it would be helpful, to see what values for shear and fall-off were detected in this specific event (e.g. in L. 216ff).
   The values of low-level shear for the observed LLJ can be found in all panels of Figure 7. We have not considered drop-off as a metric to compare within this study because it is limited to 200 m (the maximum height of the observations). We do not consider the drop-off value to be within the scope of the paper.

3. Fig. 7/ Fig.8: From the time series data in Fig. 7 and the profiles in Fig. 8, it is seen that the low-level shear, as well as the fall-off is considerably smaller for the mesoscale domains. Could you elaborate on whether the LLJ definition you applied, detects the LLJ throughout all domains and all different set-ups.
   This is true that the mesoscale domain has a higher jet nose height and so, when limiting the view to 200 m, the drop-off is much lower than the microscale domains and observations. For the simulations, the low-level jet is defined strictly by the height of maximum wind speed. An LLJ detection algorithm was used to find an event to simulate in this study, but the analysis of the detection algorithm performance is not in the scope of this paper.

4. L.285/Fig. 9: For both hub height wind speed and REWS, EMEs larger than 1 ms⁻¹ occur at times. Given that you already calculated the REWS, would it be possible to elaborate on how these differences in wind speed translate to differences in possible power production, as power changes with the cube of the velocity. I see, that your specific case shows wind speeds that are probably above rated wind speed for the turbine sizes you assume. This actually makes it a two-part comment: a) How do the EMEs and Spreads translate to lower wind speeds and b) how large is their effect on

an exemplary turbine's power production?

We appreciate the suggestion to analyze potential power production impacts and have included the following text in the manuscript:

*"Note that the wind speeds modeled are in the rated portion of most wind turbines. For reference, if we were in the cubic portion of the power curve, over-prediction of wind speeds by this amount would result in over-predictions of energy production during this period by between 3–16% for the mesoscale domains and between 15–27% on domain 3 and the LES domains (assuming wind speeds are below rated wind speed and above the cut-in speed, a performance coefficient of 0.4, and an average air density of 1.225 kg m³)."*